# Inflammasomes primarily restrict cytosolic *Salmonella* replication within human macrophages

**Marisa S Egan[1†], Emily A O'Rourke[1], Shrawan Kumar Mageswaran[2,3], Biao Zuo[3,4], Inna Martynyuk[3,4], Tabitha Demissie[1], Emma N Hunter[1], Antonia R Bass[1‡], Yi-Wei Chang[2,3], Igor E Brodsky[5], Sunny Shin[1]***

[1]Department of Microbiology, Perelman School of Medicine, University of Pennsylvania, Philadelphia, United States; [2]Department of Biochemistry and Biophysics, Perelman School of Medicine, University of Pennsylvania, Philadelphia, United States; [3]Institute of Structural Biology, Perelman School of Medicine, University of Pennsylvania, Philadelphia, United States; [4]Electron Microscopy Resource Laboratory, Department of Biochemistry & Biophysics, Perelman School of Medicine, University of Pennsylvania, Philadelphia, United States; [5]Department of Pathobiology, University of Pennsylvania School of Veterinary Medicine, Philadelphia, United States

*For correspondence:
sunshin@pennmedicine.upenn.edu

Present address: [†]Swarthmore College, Swarthmore, United States; [‡]MRL, Merck & Co, Inc, Rahway, United States

Competing interest: The authors declare that no competing interests exist.

## eLife Assessment

This paper provides **fundamental** insights into the control of *Salmonella* within human macrophages, with **convincing** evidence that *Salmonella* can replicate in the macrophage cytosol in the absence of inflammasome signaling. This paper, which improves our understanding of how the immune system fights bacterial infections, will be of broad interest to cell biologists, immunologists and microbiologists.

**Abstract** *Salmonella enterica* serovar Typhimurium is a facultative intracellular pathogen that utilizes its type III secretion systems (T3SSs) to inject virulence factors into host cells and colonize the host. In turn, a subset of cytosolic immune receptors respond to T3SS ligands by forming multimeric signaling complexes called inflammasomes, which activate caspases that induce inter-leukin-1 (IL-1) family cytokine release and an inflammatory form of cell death called pyroptosis. Human macrophages mount a multifaceted inflammasome response to *Salmonella* infection that ultimately restricts intracellular bacterial replication. However, how inflammasomes restrict *Salmonella* replication remains unknown. We find that caspase-1 is essential for mediating inflammasome responses to *Salmonella* and restricting bacterial replication within human macrophages, with caspase-4 contributing as well. We also demonstrate that the downstream pore-forming protein gasdermin D (GSDMD) and Ninjurin-1 (NINJ1), a mediator of terminal cell lysis, play a role in controlling *Salmonella* replication in human macrophages. Notably, in the absence of inflammasome responses, we observed hyperreplication of *Salmonella* within the cytosol of infected cells as well as increased bacterial replication within vacuoles, suggesting that inflammasomes control *Salmonella* replication primarily within the cytosol and also within vacuoles. These findings reveal that inflammatory caspases and pyroptotic factors mediate inflammasome responses that restrict the subcellular localization of intracellular *Salmonella* replication within human macrophages.

## Introduction

Intracellular bacterial pathogens create and maintain replicative niches within host cells. To survive inside the host, these pathogens must remodel the host cellular landscape and overcome immune defenses. *Salmonella enterica* serovar Typhimurium (*Salmonella*) is a facultative intracellular pathogen and a major cause of food-borne illness worldwide (*Majowicz et al., 2010*). Following ingestion, *Salmonella* colonizes the intestinal tract, where it can invade, replicate, and survive in host cells, including macrophages (*Crowley and Knodler, 2016*; *Santos and Bäumler, 2004*). *Salmonella* employs type III secretion systems (T3SSs) that act as molecular syringes to inject virulence factors, or effectors, into the host cell cytosol (*Agbor and McCormick, 2011*; *Crowley and Knodler, 2016*). Specifically, *Salmonella* relies on two distinct T3SSs encoded on *Salmonella* Pathogenicity Islands 1 and 2 (SPI-1 and SPI-2) to invade and replicate within host cells, respectively (*Mills et al., 1995*; *Shea et al., 1996*; *Hensel et al., 1998*; *Galan and Zhou, 2000*; *Galán, 1999a*; *Galán and Collmer, 1999b*; *Galán and Curtiss, 1989*; *Ochman et al., 1996*; *Cirillo et al., 1998*). The SPI-2 T3SS translocates effectors to facilitate biogenesis and maintenance of the *Salmonella*-containing vacuole (SCV), wherein *Salmonella* resides and replicates (*Cirillo et al., 1998*; *Takeuchi, 1967a*; *Takeuchi and Sprinz, 1967b*; *Kihlström and Latkovic, 1978*; *Garcia del Portillo et al., 1993*; *Garcia del Portillo and Finlay, 1995*; *Steele Mortimer et al., 1999*; *Hansen Wester et al., 2002*; *Steele Mortimer, 2008*; *Jennings et al., 2017*; *Brumell et al., 2002a*; *Brumell et al., 2002b*; *Beuzón et al., 2000*; *Hensel et al., 1998*). In epithelial cells, *Salmonella* escapes its vacuolar niche and hyperreplicates in the host cell cytosol (*Knodler et al., 2014a*; *Knodler et al., 2010*; *Knodler et al., 2014b*; *Malik Kale et al., 2012*). While the activity of T3SSs are essential for *Salmonella* to infect and replicate within host cells, they also inject bacterial ligands that enable innate immune sensing of *Salmonella* (*Miao et al., 2010b*; *Rayamajhi et al., 2013*; *Sun et al., 2007*; *Yang et al., 2013*).

The mammalian innate immune system detects violations of cytosolic sanctity, such as the presence of intracellular bacterial pathogens, through cytosolic pattern recognition receptors (PRRs) (*Janeway, 1989*; *Lamkanfi and Dixit, 2009*; *Medzhitov and Janeway, 2002*). A subset of these PRRs includes the nucleotide-binding domain, leucine-rich repeat (NLR) family proteins. NLRs respond to their cognate stimuli by oligomerizing and inducing the assembly of multiprotein complexes termed inflammasomes, which activate inflammatory caspases (*Broz and Dixit, 2016*; *Lamkanfi and Dixit, 2009*; *Lamkanfi and Dixit, 2014*; *Martinon et al., 2002*). Canonical inflammasomes recruit and activate caspase-1, which cleaves and activates pro-inflammatory IL-1 family cytokines and the pore-forming protein gasdermin D (GSDMD) (*Kuida et al., 1995*; *Li et al., 1995*; *Agard et al., 2010*; *Thornberry et al., 1992*; *Shi et al., 2015*; *Kayagaki et al., 2015*). Alternatively, noncanonical inflammasomes are formed by caspase-11 in mice and two orthologs in humans, caspase-4 and caspase-5, in response to cytosolic lipopolysaccharide (LPS) (*Casson et al., 2015*; *Hagar et al., 2013*; *Kayagaki et al., 2011*; *Kayagaki et al., 2013*; *Lagrange et al., 2018*; *Schmid Burgk et al., 2015*; *Shi et al., 2014*). These caspases directly process and activate GSDMD (*Kayagaki et al., 2015*; *Shi et al., 2015*). Liberated GSDMD N-terminal fragments oligomerize to create pores in the host plasma membrane, through which IL-1 family cytokines and other alarmins are released to promote a lytic form of inflammatory cell death termed pyroptosis (*Agard et al., 2010*; *Ding et al., 2016*; *Kayagaki et al., 2015*; *Shi et al., 2015*).

*Salmonella* activates several inflammasomes in human macrophages. One such inflammasome is the NLR family, apoptosis inhibitory protein (NAIP)/NLR family, CARD domain-containing protein 4 (NLRC4) inflammasome (*Bierschenk et al., 2019*; *Gram et al., 2021*; *Naseer et al., 2022a*). NAIP detects the cytosolic presence of T3SS structural components and flagellin (*Grandjean et al., 2017*; *Kofoed and Vance, 2011*; *Kortmann et al., 2015*; *Miao et al., 2010b*; *Molofsky et al., 2006*; *Rauch et al., 2016*; *Rayamajhi et al., 2013*; *Ren et al., 2006*; *Reyes Ruiz et al., 2017*; *Sun et al., 2007*; *Yang et al., 2013*; *Zhao et al., 2011*; *Zhao et al., 2016*). Upon ligand recognition, NAIP recruits NLRC4, which oligomerizes to form the active NAIP/NLRC4 inflammasome (*Diebolder et al., 2015*; *Hu et al., 2015*; *Zhang et al., 2015*). *Salmonella* also activates the NLR pyrin domain-containing protein 3 (NLRP3) inflammasome and the noncanonical caspase-4/5 inflammasome in human macrophages (*Bierschenk et al., 2019*; *Casson et al., 2015*; *Gram et al., 2021*; *Naseer et al., 2022a*). NLRP3 responds to diverse stimuli, including ionic fluxes as a result of host plasma membrane damage (*Franchi et al., 2007*; *Hornung et al., 2008*; *Mariathasan et al., 2006*; *Muñoz Planillo et al., 2013*; *Perregaux and Gabel, 1994*), and can be secondarily activated by the noncanonical inflammasome

(*Baker et al., 2015*; *Casson et al., 2015*; *Kayagaki et al., 2013*; *Pilla et al., 2014*; *Rathinam et al., 2012*; *Rühl and Broz, 2015*; *Schmid Burgk et al., 2015*; *Shi et al., 2014*).

Inflammasome activation is critical for host defense against *Salmonella*. In mice, the NAIP/NLRC4 inflammasome is required to control *Salmonella* infection (*Carvalho et al., 2012*; *Franchi et al., 2012*; *Hausmann et al., 2020*; *Miao et al., 2006*; *Miao et al., 2010b*; *Rauch et al., 2016*; *Rauch et al., 2017*; *Sellin et al., 2014*; *Zhao et al., 2016*). In murine intestinal epithelial cells (IECs), NAIP/NLRC4 inflammasome activation triggers pyroptosis and expulsion of infected cells, and is both necessary and sufficient in IECs to restrict *Salmonella* replication and prevent bacterial dissemination to distal organ sites (*Hausmann et al., 2020*; *Rauch et al., 2017*; *Sellin et al., 2014*). Unlike murine IECs, human epithelial cells do not rely on NAIP/NLRC4 or caspase-1, but instead rely on caspase-4 to control *Salmonella* replication, in part through pyroptosis and cell extrusion (*Holly et al., 2020*; *Knodler et al., 2014a*; *Knodler et al., 2010*; *Naseer et al., 2022b*). Moreover, in murine macrophages, inflammasome activation controls the replication of a mutant strain of *Salmonella* that aberrantly invades the cytosol, but only slightly limits the replication of wild-type (WT) *Salmonella* (*Man et al., 2014a*; *Thurston et al., 2016*). In contrast, we found that human macrophages rely on NAIP/NLRC4- and NLRP3-dependent inflammasome responses to control intracellular WT *Salmonella* replication (*Naseer et al., 2022a*). However, how inflammasome signaling restricts *Salmonella* replication in human macrophages remains unknown.

In this study, we show that caspase-1 is required to restrict intracellular *Salmonella* replication early during infection in human macrophages, and that caspase-4 enables the restriction of *Salmonella* later during infection. We find that the cell lysis mediators GSDMD and ninjurin-1 (NINJ1) also contribute to bacterial restriction. Importantly, we observed that while human macrophages unable to undergo inflammasome responses showed slightly elevated bacterial replication within SCVs, they became permissive to hyperreplication of *Salmonella* within the cytosolic compartment. Thus, inflammasome activation appears to preferentially restrict cytosolic *Salmonella* replication. Our results offer insight into how inflammasomes control *Salmonella* in human macrophages and restrict the distinct intracellular spatial niches that *Salmonella* occupies in these cells.

## Results

### Caspase-1 promotes the control of *Salmonella* replication within human macrophages

Human macrophages undergo NAIP/NLRC4- and NLRP3-dependent inflammasome activation during *Salmonella* infection (*Bierschenk et al., 2019*; *Gram et al., 2021*; *Naseer et al., 2022a*), which restricts intracellular *Salmonella* replication (*Naseer et al., 2022a*). Nonetheless, how inflammasome activation controls *Salmonella* replication in human macrophages remains unclear. Inflammasomes recruit and activate caspase-1 in both murine and human cells (*Franchi et al., 2006*; *Man et al., 2014b*; *Mariathasan et al., 2004*; *Miao et al., 2006*; *Ross et al., 2022*; *Zamboni et al., 2006*), and caspase-1 promotes the control of *Salmonella* both in vivo and in vitro (*Broz et al., 2010*; *Broz et al., 2012*; *Crowley et al., 2020*; *Hausmann et al., 2020*; *Holly et al., 2020*; *Lara Tejero et al., 2006*; *Miao et al., 2010a*; *Rauch et al., 2017*; *Raupach et al., 2006*; *Sellin et al., 2014*; *Thurston et al., 2016*). Thus, we sought to test whether caspase-1 restricts *Salmonella* replication specifically in human macrophages.

To first interrogate the impact of caspase activity on *Salmonella* replication in human macrophages, we primed macrophages derived from the human monocytic cell line, THP-1, with the TLR1/2 agonist Pam3CSK4 to upregulate the expression of inflammasome components. Then, we pretreated the cells with Ac-YVAD-cmk (YVAD), a chemical inhibitor of caspase-1 activity, or Z-VAD-FMK (ZVAD), a pan-caspase inhibitor. Upon infection with wild-type (WT) *Salmonella*, WT THP-1 cells pretreated with either YVAD or ZVAD had significantly reduced levels of IL-1β release and cell death compared to infected WT cells pretreated with the vehicle control (*Figure 1—figure supplement 1A–B*). Since pretreatment with YVAD largely phenocopies ZVAD, these data suggest that caspase-1 is the primary caspase that responds to *Salmonella* infection in THP-1 cells. We next assessed intracellular *Salmonella* burdens by determining bacterial colony-forming units (CFUs). The increase in bacterial CFUs was significantly higher in WT cells pretreated either YVAD or ZVAD compared to cells pretreated with DMSO (*Figure 1—figure supplement 1C*). Microscopic analysis revealed that WT cells treated with

either YVAD or ZVAD prior to infection harbored significantly higher intracellular *Salmonella* burdens compared to cells pretreated with DMSO (*Figure 1—figure supplement 1D*). Overall, these results suggest that caspase-1 activity primarily controls *Salmonella* burdens in human macrophages.

Next, to genetically test the requirement of caspase-1, we used two independent *CASP1⁻/⁻* THP-1 single-cell clones generated through CRISPR/Cas9-mediated deletion (*Okondo et al., 2017*). In agreement with previous reports (*Bierschenk et al., 2019*; *Gram et al., 2021*; *Naseer et al., 2022a*), WT THP-1 cells primed with Pam3CSK4 and then infected with WT *Salmonella* exhibited high levels of IL-1β, IL-18, and IL-1α release as well as cell death at 6 hpi (*Figure 1A–B*; *Figure 1—figure supplement 2A–B*). In contrast, Pam3CSK4-primed *CASP1⁻/⁻* THP-1 cells released negligible levels of inflammasome-dependent cytokines and did not undergo substantial cell death upon infection (*Figure 1A–B*; *Figure 1—figure supplement 2A–B*). Infected WT and *CASP1⁻/⁻* THP-1 cells released similar levels of the inflammasome-independent cytokine TNF-α (*Figure 1—figure supplement 2C*). These results indicate that caspase-1 is required for inflammasome responses to *Salmonella* infection in human macrophages.

We next tested whether caspase-1 is required to restrict *Salmonella* replication in human macrophages. At 1 hr post-infection (hpi), we did not observe any significant differences in bacterial uptake between Pam3CSK4-primed WT and *CASP1⁻/⁻* THP-1 cells (*Figure 1—figure supplement 2D*). However, at 6 hpi, *CASP1⁻/⁻* cells harbored significantly higher bacterial burdens and a significant fold-increase in bacterial CFUs compared to WT cells (*Figure 1C*; *Figure 1—figure supplement 2E*). To assess intracellular bacterial burdens on a single-cell level, we quantified the number of WT *Salmonella* per cell by microscopy. WT cells contained relatively low numbers of *Salmonella* on average (~5 bacteria per cell), while *CASP1⁻/⁻* cells harbored significantly higher numbers of *Salmonella* on average (~30 bacteria per cell) at 6 hpi (*Figure 1D–E*). Overall, these data indicate that caspase-1 limits intracellular *Salmonella* replication in human macrophages.

We next examined whether caspase-1-mediated control of *Salmonella* replication in human macrophages requires TLR1/2 priming. At 1 hpi, we did not observe any significant differences in bacterial uptake between unprimed WT and *CASP1⁻/⁻* THP-1 cells (*Figure 1—figure supplement 3A*). However, at 6 hpi, unprimed *CASP1⁻/⁻* cells harbored significantly higher bacterial burdens and a significant fold-increase in bacterial CFUs compared to unprimed WT cells (*Figure 1—figure supplement 3B–C*). Since unprimed conditions phenocopy TLR1/2-primed conditions (*Figure 1C*; *Figure 1—figure supplement 2D–E*), these data indicate that caspase-1 promotes the restriction of *Salmonella* replication within human macrophages independent of TLR1/2 priming.

In the previous experiments, we infected THP-1 cells with *Salmonella* grown under SPI-1-inducing conditions. So, we subsequently sought to determine whether *Salmonella* grown to stationary phase similarly replicated in human macrophages. At 1 hpi and at 6 hpi, we did not observe any significant differences in bacterial uptake between WT and *CASP1⁻/⁻* THP-1 cells infected with stationary phase *Salmonella* (*Figure 1—figure supplement 4A–B*). Strikingly, we did not observe any fold-change in the bacterial CFUs in both WT and *CASP1⁻/⁻* THP-1 cells (*Figure 1—figure supplement 4C*). These data indicate that by 6 hr post-infection, *Salmonella* do not replicate efficiently in human macrophages unless grown under SPI-1-inducing conditions (*Figure 1C*; *Figure 1—figure supplement 2D–E*).

Next, we sought to address the possibility that the caspase-1-mediated control of *Salmonella* replication we observed was due to an experimental artifact of gentamicin entering WT macrophages through caspase-1-dependent gasdermin D pores and subsequently killing intracellular bacteria. At 30 min post-infection, THP-1 cells were treated with a lower dose of 25 μg/ml of gentamicin to kill extracellular bacteria. At 1 hr post-infection (hpi), the cells were washed to remove the gentamicin, and then the media was replaced with fresh media containing no gentamicin. Alternatively, we treated cells with our standard bactericidal concentration of gentamicin (100 μg/ml), washed the cells to remove the gentamicin, and then replaced the media with fresh media containing a bacteriostatic concentration of gentamicin (10 μg/ml). At 1 hpi, we did not observe any significant differences in bacterial uptake between WT and *CASP1⁻/⁻* THP-1 cells treated with 25 μg/ml of gentamicin or treated with 100 μg/ml of gentamicin (*Figure 1—figure supplement 5A*). However, at 6 hpi, regardless of the presence or absence of gentamicin, *CASP1⁻/⁻* cells harbored significantly higher bacterial burdens and a significant fold-increase in bacterial CFUs compared to WT cells (*Figure 1—figure supplement 5B–C*). Altogether, these data indicate that gentamicin does not contribute to caspase-1-mediated restriction of *Salmonella* replication in human macrophages.

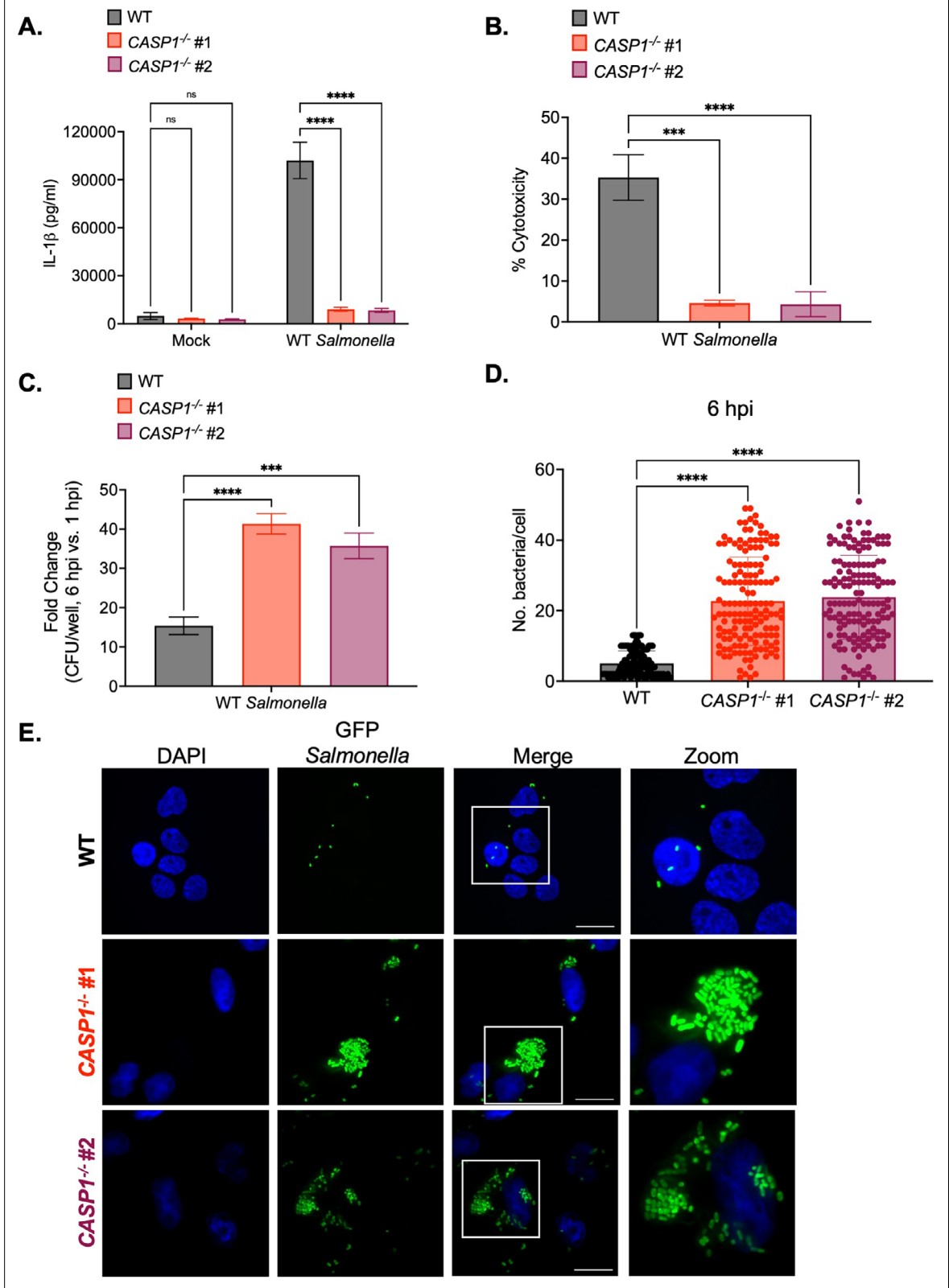

**Figure 1.** Caspase-1 promotes the control of *Salmonella* replication within human macrophages. Wild-type (WT) and two independent clones of *CASP1*-/- THP-1 monocyte-derived macrophages were primed with 100 ng/mL Pam3CSK4 for 16 hr. Cells were then infected with PBS (Mock), WT *S.* Typhimurium (**A, B, C**), or WT *S.* Typhimurium constitutively expressing GFP (**D,E**) at an MOI = 20. (**A**) Release of IL-1β into the supernatant was measured by ELISA at 6 hpi. (**B**) Cell death (% cytotoxicity) was measured by LDH release assay and normalized to mock-infected cells at 6 hpi. (**C**) Cells

*Figure 1 continued on next page*

*Figure 1 continued*

were lysed at 1 hpi and 6 hpi, and bacteria were subsequently plated to calculate colony-forming units (CFUs). Fold-change in CFU/well was calculated. (**D, E**) Cells were fixed at 6 hpi and stained for DAPI to label DNA (blue). (**D**) The number of bacteria per cell at 6 hpi was scored by fluorescence microscopy. Each dot represents one infected cell. 150 infected cells were scored for each genotype. (**E**) Representative images are shown. Scale bar represents 10 µm. Bars represent the mean for each genotype, and error bars represent the standard deviation of triplicate wells from one experiment (**A, B, C, D**). ***p<0.001, ****p<0.0001 by Dunnett's multiple comparisons test (**A, B, C, D**). Data shown are representative of at least three independent experiments.

The online version of this article includes the following source data and figure supplement(s) for figure 1:

**Source data 1.** The numerical source data that corresponds to *Figure 1*.

**Figure supplement 1.** Caspase activity promotes the control of *Salmonella* replication within human macrophages.

**Figure supplement 1—source data 1.** The numerical source data that corresponds to *Figure 1—figure supplement 1*.

**Figure supplement 2.** Caspase-1 promotes the control of *Salmonella* replication within human macrophages.

**Figure supplement 2—source data 1.** The numerical source data that corresponds to *Figure 1—figure supplement 2*.

**Figure supplement 3.** Caspase-1 promotes the control of *Salmonella* replication within unprimed human macrophages.

**Figure supplement 3—source data 1.** The numerical source data that corresponds to *Figure 1—figure supplement 3*.

**Figure supplement 4.** Stationary phase *Salmonella* does not replicate effectively in human macrophages.

**Figure supplement 4—source data 1.** The numerical source data that corresponds to *Figure 1—figure supplement 4*.

**Figure supplement 5.** Gentamicin does not contribute to the control of *Salmonella* replication in human macrophages.

**Figure supplement 5—source data 1.** The numerical source data that corresponds to *Figure 1—figure supplement 5*.

## Caspase-4 contributes to the control of *Salmonella* replication within human macrophages later during infection

Caspase-4 contributes to inflammasome responses during *Salmonella* infection of THP-1 macrophages and primary human macrophages (*Casson et al., 2015*; *Naseer et al., 2022a*), and caspases-4/5 are required for inflammasome responses to *Salmonella* in THP-1 monocytes (*Baker et al., 2015*). In human intestinal epithelial cells (IECs), caspase-4 drives inflammasome responses during *Salmonella* infection and limits intracellular bacterial replication (*Holly et al., 2020*; *Knodler et al., 2014a*; *Naseer et al., 2022b*). However, whether caspase-4 contributes to restriction of intracellular *Salmonella* replication within human macrophages is unclear.

To genetically test the contribution of caspase-4 during *Salmonella* infection in THP-1 macrophages, we used CRISPR/Cas9 to disrupt the *CASP4* gene. We selected and sequence-validated two independent *CASP4*[-/-] THP-1 single-cell clones (*Figure 2—figure supplement 1*). We observed a slight decrease in secreted IL-1β levels in *CASP4*[-/-] THP-1 macrophages infected with WT *Salmonella* compared to infected WT THP-1 macrophages at 6 hpi (*Figure 2—figure supplement 2A*). We also observed a slight decrease in cell death at 6 hpi in infected *CASP4*[-/-] clone #2 compared to infected WT cells, whereas cytotoxicity levels were not significantly affected in infected *CASP4*[-/-] clone #6 (*Figure 2—figure supplement 2B*). However, at 24 hpi, we observed a significant decrease in IL-1 release and cell death in both *CASP4*[-/-] clones infected with WT *Salmonella* compared to infected WT cells (*Figure 2A–B*; *Figure 2—figure supplement 3A–B*). WT and *CASP4*[-/-] cells also released similar levels of the inflammasome-independent cytokine TNF-α upon infection with WT *Salmonella* at 24 hpi (*Figure 2—figure supplement 3C*). Overall, these data indicate that caspase-4 contributes to inflammasome responses in THP-1 macrophages later during *Salmonella* infection.

Since caspase-4 restricts *Salmonella* replication in human epithelial cells (*Holly et al., 2020*; *Knodler et al., 2014a*; *Naseer et al., 2022b*), we asked whether caspase-4 contributes to the control of *Salmonella* replication in human macrophages as well. Upon examination of the fold-change in bacterial CFUs at 6 hpi, we did not observe any significant differences between WT and *CASP4*[-/-] THP-1 cells (*Figure 2—figure supplement 2C*). However, at 24 hpi, *CASP4*[-/-] cells harbored higher bacterial burdens and a significant fold-increase in bacterial CFUs compared to WT cells (*Figure 2C*; *Figure 2—figure supplement 3D*). To confirm these findings, we enumerated the amount of WT *Salmonella* per cell by microscopy. At 6 hpi, WT and *CASP4*[-/-] THP-1 cells contained comparable numbers of *Salmonella* per cell (*Figure 2—figure supplement 2D–E*). In contrast, at 24 hpi, *CASP4*[-/-] cells harbored significantly higher burdens of *Salmonella* per cell on average (~20 bacteria per cell)

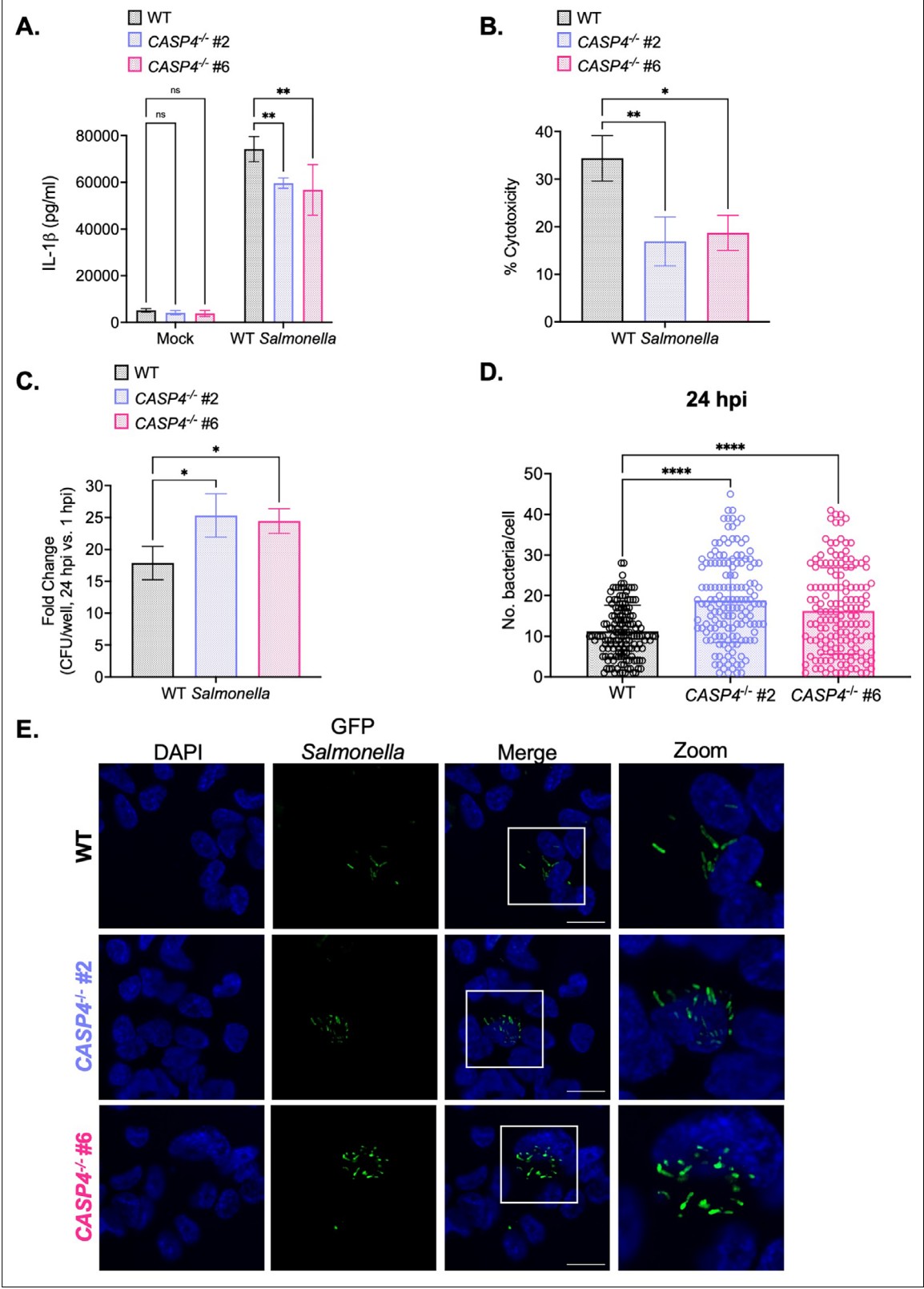

**Figure 2.** Caspase-4 contributes to the control of *Salmonella* replication within human macrophages later during infection. Wild-type (WT) and two independent clones of *CASP4⁻/⁻* THP-1 monocyte-derived macrophages were primed with 100 ng/mL Pam3CSK4 for 16 hr. Cells were then infected with PBS (Mock), WT *S.* Typhimurium (**A, B, C**), or WT *S.* Typhimurium constitutively expressing GFP (**D, E**) at an MOI = 20. (**A**) Release of IL-1β into the supernatant was measured by ELISA at 24 hpi. (**B**) Cell death (% cytotoxicity) was measured by lactate dehydrogenase (LDH) release assay and

*Figure 2 continued on next page*

*Figure 2 continued*

normalized to mock-infected cells at 24 hpi. (**C**) Cells were lysed at 1 hpi and 24 hpi, and bacteria were subsequently plated to calculate colony-forming units (CFUs). Fold-change in CFU/well was calculated. (**D, E**) Cells were fixed at 24 hpi and stained for DAPI to label DNA (blue). (**D**) The number of bacteria per cell at 24 hpi was scored by fluorescence microscopy. Each dot represents one infected cell. 150 infected cells were scored for each genotype. (**E**) Representative images are shown. Scale bar represents 10 μm. Bars represent the mean for each genotype, and error bars represent the standard deviation of triplicate wells from one experiment (**A, B, C, D**). ns – not significant, *p<0.05 by Dunnett's multiple comparisons test (**A, B, C, D**). Data shown are representative of at least three independent experiments.

The online version of this article includes the following source data and figure supplement(s) for figure 2:

**Source data 1.** The numerical source data that corresponds to *Figure 2*.

**Figure supplement 1.** Validation of *CASP4<sup>-/-</sup>* THP-1 clones generated with CRISPR/Cas9-mediated genome editing.

**Figure supplement 1—source data 1.** The original blots for *Figure 2—figure supplement 1F*.

**Figure supplement 2.** Caspase-4 is dispensable for the control of *Salmonella* replication within human macrophages early during infection.

**Figure supplement 2—source data 1.** The numerical source data that corresponds to *Figure 2—figure supplement 2*.

**Figure supplement 3.** Caspase-4 contributes to the control of *Salmonella* replication within human macrophages later during infection.

**Figure supplement 3—source data 1.** The numerical source data that corresponds to *Figure 2—figure supplement 3*.

---

compared to WT cells (~10 bacteria per cell) (*Figure 2D–E*). Altogether, these results suggest that caspase-4 plays a larger role in controlling *Salmonella* burdens later during infection.

## GSDMD promotes the control of *Salmonella* replication within human macrophages

Inflammasome activation triggers a lytic form of cell death known as pyroptosis (*Lamkanfi and Dixit, 2014*). Death of the infected host cell eliminates *Salmonella*'s intracellular replicative niche and thus, may contribute to restricting intracellular bacterial replication. Upon its cleavage by inflammatory caspases, GSDMD forms pores in the host plasma membrane, resulting in pyroptosis (*Ding et al., 2016*; *Kayagaki et al., 2015*; *Shi et al., 2015*). Whether GSDMD contributes to the control of *Salmonella* replication in human macrophages is unknown.

First, we asked whether GSDMD pore formation plays a role in controlling *Salmonella* replication in human macrophages. To do this, we pretreated THP-1 macrophages and primary human monocyte-derived macrophages (hMDMs) with the chemical inhibitor disulfiram. Disulfiram prevents cleaved GSDMD from inserting into the host plasma membrane, thereby limiting GSDMD-mediated pore formation (*Hu et al., 2020*). Disulfiram treatment led to the loss of IL-1β release and cytotoxicity in macrophages infected with WT *Salmonella*, compared to treatment with the vehicle control, DMSO (*Figure 3—figure supplement 1A–B, D–E*). Next, we examined the effect of GSDMD-mediated pore formation on intracellular *Salmonella* burdens. The fold-increase in bacterial CFUs at 6 hpi was significantly higher in cells treated with disulfiram compared to cells treated with DMSO (*Figure 3—figure supplement 1C, F*). Collectively, these results suggest that GSDMD-mediated pore formation promotes the restriction of intracellular *Salmonella* replication.

Consistent with our findings with disulfiram, we observed a significant decrease in IL-1 and IL-18 release in *GSDMD<sup>-/-</sup>* THP-1 macrophages (*Okondo et al., 2017*; *Taabazuing et al., 2017*) compared to WT THP-1 macrophages following *Salmonella* infection (*Figure 3A*; *Figure 3—figure supplement 2A–B*). Interestingly, loss of GSDMD did not completely abrogate the release of IL-1 and IL-18, suggesting that there is also GSDMD-independent IL-1 and IL-18 release (*Figure 3A*; *Figure 3—figure supplement 2A–B*). Infected WT and *GSDMD<sup>-/-</sup>* THP-1 cells secreted similar levels of the inflammasome-independent cytokine TNF-α (*Figure 3—figure supplement 2C*). Importantly, infected *GSDMD<sup>-/-</sup>* cells failed to undergo substantial cell death, in contrast to infected WT cells, indicating that GSDMD facilitates cell death during *Salmonella* infection (*Figure 3B*). Overall, these results indicate that GSDMD plays an important role in mediating inflammasome responses during *Salmonella* infection of human macrophages.

We next examined whether GSDMD controls intracellular *Salmonella* replication in human macrophages. Notably, while there were no differences in bacterial uptake between WT and *GSDMD<sup>-/-</sup>* cells at 1 hpi (*Figure 3—figure supplement 2D*), we did observe significantly higher bacterial CFUs in *GSDMD<sup>-/-</sup>* cells compared to WT cells at 6 hpi (*Figure 3—figure supplement 2E*). Moreover, the fold-increase in bacterial CFUs at 6 hpi was significantly higher in *GSDMD<sup>-/-</sup>* cells than in WT cells

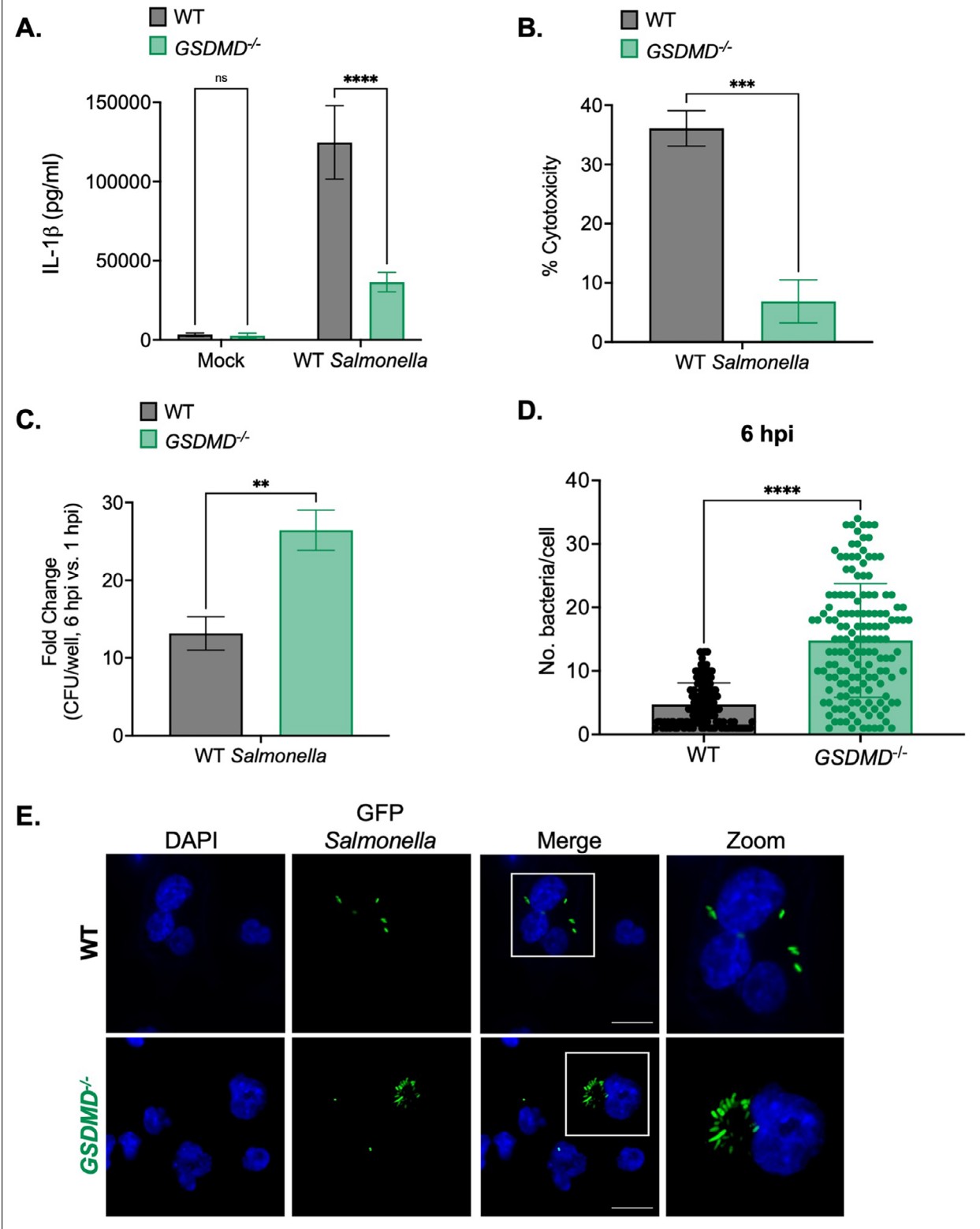

**Figure 3.** Gasdermin D (GSDMD) promotes the control of *Salmonella* replication within human macrophages. Wild-type (WT) and *GSDMD$^{-/-}$* THP-1 monocyte-derived macrophages were primed with 100 ng/mL Pam3CSK4 for 16 hr. Cells were then infected with PBS (Mock), WT *S*. Typhimurium (**A, B, C**), or WT *S*. Typhimurium constitutively expressing GFP (**D,E**) at an MOI = 20. (**A**) Release of IL-1β into the supernatant was measured by ELISA at 6 hpi. (**B**) Cell death (% cytotoxicity) was measured by LDH release assay and normalized to mock-infected cells at 6 hpi. (**C**) Cells were lysed at 1 hpi and 6 hpi, and bacteria were subsequently plated to calculate colony-forming units (CFUs). Fold-change in CFU/well was calculated. (**D, E**) Cells were fixed at 6

*Figure 3 continued on next page*

*Figure 3 continued*

hpi and stained for DAPI to label DNA (blue). (**D**) The number of bacteria per cell at 6 hpi was scored by fluorescence microscopy. Each dot represents one infected cell. 150 infected cells were scored for each genotype. (**E**) Representative images are shown. Scale bar represents 10 µm. Bars represent the mean for each genotype, and error bars represent the standard deviation of triplicate wells from one experiment (**A, B, C, D**). **p<0.01, ***p<0.001, ****p<0.0001 by Šídák's multiple comparisons test (**A**) or by unpaired t-test (**B, C, D**). Data shown are representative of at least three independent experiments.

The online version of this article includes the following source data and figure supplement(s) for figure 3:

**Source data 1.** The numerical source data that corresponds to *Figure 3*.

**Figure supplement 1.** Gasdermin D (GSDMD)-mediated pore formation promotes the control of *Salmonella* replication within human macrophages.

**Figure supplement 1—source data 1.** The numerical source data that corresponds to *Figure 3—figure supplement 1*.

**Figure supplement 2.** Gasdermin D (GSDMD) promotes the control of *Salmonella* replication within human macrophages.

**Figure supplement 2—source data 1.** The numerical source data that corresponds to *Figure 3—figure supplement 2*.

(*Figure 3C*). Next, we used microscopy to further interrogate intracellular *Salmonella* burdens. As expected, WT cells contained low numbers of *Salmonella* per cell (~5 bacteria per cell) (*Figure 3D–E*). However, *GSDMD*[-/-] cells harbored significantly higher numbers of *Salmonella* per cell on average (~20 bacteria per cell) at 6 hpi (*Figure 3D–E*). Altogether, these data indicate that GSDMD promotes the control of *Salmonella* replication in human macrophages.

## NINJ1 contributes to the control of *Salmonella* replication within human macrophages

While GSDMD pores release a subset of molecules, including IL-1 family cytokines, the extensive release of cellular contents is mediated by plasma membrane rupture (*Ding et al., 2016*; *Ruan et al., 2018*). Recently, NINJ1 was identified as an executioner of terminal cell lysis (*Bjanes et al., 2021*; *Borges et al., 2022*; *Degen et al., 2023*; *Kayagaki et al., 2021*; *Kayagaki et al., 2023*). NINJ1 is a transmembrane protein that oligomerizes in the host plasma membrane to induce lytic rupture of the cell after the initiation of pyroptosis and other forms of regulated cell death (*David et al., 2024*; *Degen et al., 2023*; *Kayagaki et al., 2021*). Cells can be protected from plasma membrane rupture through treatment with the amino acid glycine (*Fink and Cookson, 2006*; *Frank et al., 2000*; *Heilig et al., 2018*; *Verhoef et al., 2005*), which interferes with the clustering of NINJ1 (*Borges et al., 2022*). Thus, upon incubation with glycine, cells can still form GSDMD pores but cannot undergo terminal cell lysis (*Fink and Cookson, 2006*; *Heilig et al., 2018*; *Tsuchiya et al., 2021*; *Verhoef et al., 2005*). Whether NINJ1-dependent cell lysis contributes to control of *Salmonella* replication in human macrophages remains unknown.

First, to determine whether glycine exerts a cytoprotective effect on THP-1 macrophages, we pretreated WT and *NAIP*[-/-] THP-1 cells with glycine prior to infection with WT *Salmonella* and assayed for downstream inflammasome responses. Notably, WT and *NAIP*[-/-] cells pretreated with glycine exhibited significantly decreased cell death and a small defect in IL-1β release following infection compared to infected cells treated with the vehicle control (*Figure 4—figure supplement 1A–B*). Release of the inflammasome-independent cytokine TNF-α was unaffected by glycine treatment (*Figure 4—figure supplement 1C*). Altogether, these results indicate that glycine prevents cell lysis and limits IL-1 release in THP-1 macrophages upon *Salmonella* infection.

Given glycine's cytoprotective effect on infected THP-1 macrophages, we next sought to determine whether glycine impacts intracellular *Salmonella* burdens in human macrophages. THP-1 macrophages pretreated with glycine retained significantly higher intracellular bacterial burdens at 6 hpi compared to cells given the vehicle control (*Figure 4—figure supplement 1D*). We did not detect any significant differences in bacterial uptake (*Figure 4—figure supplement 1E*). Moreover, WT and *NAIP*[-/-] THP-1 cells pretreated with glycine exhibited a greater increase in *Salmonella* CFUs compared to cells treated with the vehicle control (*Figure 4—figure supplement 1F*), suggesting that cytoprotection by glycine interferes with the intracellular control of *Salmonella* in human macrophages.

Next, to genetically test the role of NINJ1, we transfected WT THP-1 macrophages with small interfering RNA (siRNA) targeting *NINJ1*, which led to robust knockdown of *NINJ1* ranging from 77% to 85%, or control scrambled siRNA. Knockdown of *NINJ1* resulted in a significant decrease, yet not complete abrogation, of IL-1β secretion and cytotoxicity in infected cells, in contrast to control siRNA

treatment (*Figure 4A–B*). Collectively, these data imply a critical role for NINJ1 in contributing to IL-1 release and cell death during *Salmonella* infection in human macrophages.

Notably, while *NINJ1* knockdown did not affect bacterial uptake at 1 hpi (*Figure 4C*), cells treated with *NINJ1* siRNA contained higher intracellular bacterial burdens than cells treated with control siRNA at 6 hpi (*Figure 4D*). Moreover, there was a greater fold-increase in bacterial CFUs at 6 hpi in *NINJ1* siRNA-treated cells compared to control siRNA-treated cells (*Figure 4E*). Together, these findings indicate that NINJ1 contributes to intracellular bacterial control.

## Inflammasome activation primarily controls cytosolic *Salmonella* replication in human macrophages

In epithelial cells, WT *Salmonella* can enter the cytosol and replicate in both vacuolar and cytosolic compartments, specifically hyperreplicating in the cytosol (*Knodler et al., 2014a*; *Knodler et al., 2010*; *Knodler et al., 2014b*; *Malik Kale et al., 2012*). However, in murine macrophages, WT *Salmonella* appears to replicate exclusively in vacuoles (*Beuzón et al., 2002*; *Thurston et al., 2016*). *Salmonella* lacking the SPI-2 effector SifA (Δ*sifA*), which is required for SCV membrane stability, frequently enter the cytosol and can replicate within epithelial cells, but cannot replicate efficiently in murine macrophages (*Beuzón et al., 2000*; *Beuzón et al., 2002*; *Brumell et al., 2002b*; *Thurston et al., 2016*). Whether *Salmonella* replicates within vacuoles or the cytosol of human macrophages remains largely unknown. One study suggested that a small but significant proportion of *Salmonella* are exposed to the cytosol in THP-1 macrophages (*Fisch et al., 2020*). Our recently published data and current findings indicate that when human macrophages fail to undergo robust inflammasome responses, *Salmonella* hyperreplicates to large numbers within these cells (*Naseer et al., 2022a*). The hyperreplication that we observed in human macrophages was reminiscent of previous reports describing *Salmonella* hyperreplication within the cytosol of IECs (*Knodler et al., 2014a*; *Knodler et al., 2010*; *Knodler et al., 2014b*; *Malik Kale et al., 2012*). Thus, we hypothesized that inflammasome activation limits *Salmonella* replication in both SCVs and the host cell cytosol, and that in the absence of inflammasome responses, *Salmonella* hyperreplicates to large numbers within the host cell cytosol and also exhibits increased replication within SCVs.

First, to test whether inflammasome activation restricts the ability of *Salmonella* to replicate in vacuolar and cytosolic compartments of human macrophages, we used a chloroquine (CHQ) resistance assay. As a weak base, CHQ accumulates in endosomal compartments, including the SCV, without entering the cytosol of host cells (*Klein et al., 2017b*; *Knodler et al., 2014b*; *Steinberg, 1994*). Thus, vacuolar bacteria are CHQ-sensitive, while cytosolic bacteria are CHQ-resistant (*Knodler et al., 2014b*). We infected THP-1 macrophages with WT *Salmonella* and treated the infected cells with CHQ. Then, we determined the bacterial CFUs at 2 hpi and 6 hpi, quantifying the numbers of vacuolar bacteria (CHQ-sensitive) and cytosolic bacteria (CHQ-resistant). At 2 hpi, we did not observe any significant differences in vacuolar or cytosolic *Salmonella* between WT and *CASP1*[-/-] THP-1 cells (*Figure 5A*). We found that WT THP-1 cells contained mostly vacuolar *Salmonella* while also retaining some cytosolic *Salmonella* at 6 hpi (*Figure 5B*), suggesting that a subset of *Salmonella* dwells in the cytosol of THP-1 macrophages, in agreement with a previous study (*Fisch et al., 2020*). In *NAIP*[-/-] and *CASP1*[-/-] THP-1 cells, we observed slightly higher burdens of vacuolar *Salmonella* compared to WT cells (*Figure 5B*). Strikingly, we also observed significantly larger numbers of cytosolic *Salmonella* in *NAIP*[-/-] and *CASP1*[-/-] cells compared to WT cells (*Figure 5B*). These data indicate that inflammasome signaling restricts *Salmonella* primarily within the host cell cytosol but also within SCVs in human macrophages.

The SPI-1 T3SS can damage the SCV (*Roy et al., 2004*), and the SPI-1 T3SS and its effectors contribute to *Salmonella* escape from the SCV and replication in the cytosol of epithelial cells (*Chong et al., 2019*; *Klein et al., 2017a*; *Knodler et al., 2014b*; *Röder and Hensel, 2020*). So, we next assessed whether the SPI-1 T3SS was required for the cytosolic exposure of *Salmonella* in human macrophages. We infected WT and *CASP1*[-/-] THP-1 macrophages with *Salmonella* lacking the SPI-1 T3SS translocon protein SipB (Δ*sipB*), thus preventing SPI-1 T3SS effector translocation into host cells. While we observed a significant increase in the amount of vacuolar and cytosolic Δ*sipB Salmonella* in *CASP1*[-/-] cells compared to WT cells at 6 hpi, the numbers of cytosolic Δ*sipB* were significantly lower than vacuolar Δ*sipB* in both WT and *CASP1*[-/-] cells (*Figure 5C*). We also observed lower cytosolic Δ*sipB Salmonella* burdens compared to WT *Salmonella* in both WT and *CASP1*[-/-] THP-1 cells at 6 hpi

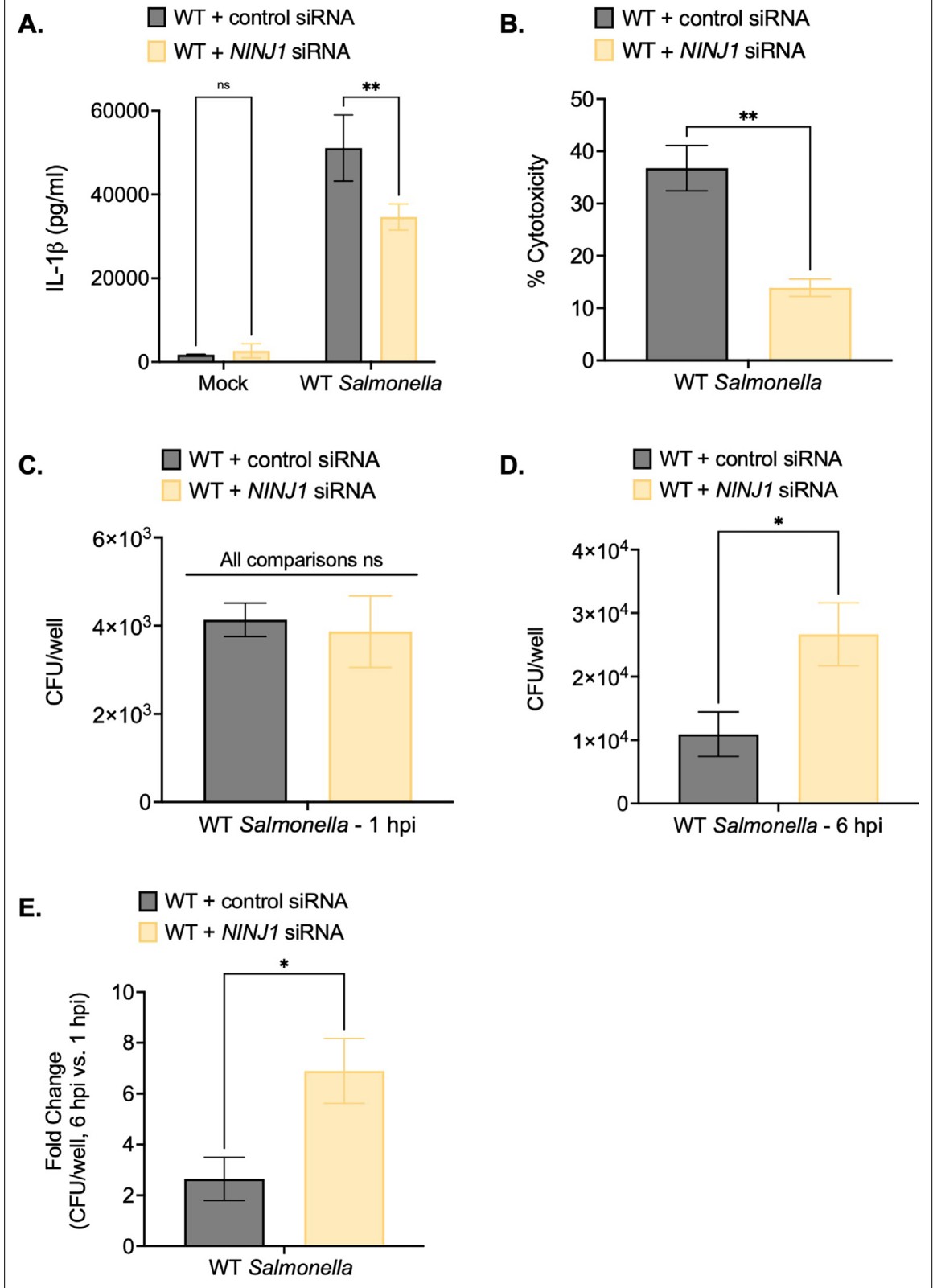

**Figure 4.** Ninjurin-1 (NINJ1) contributes to the control of *Salmonella* replication within human macrophages. Wild-type (WT) THP-1 monocyte-derived macrophages were treated with control scrambled siRNA or siRNA targeting *NINJ1* for 72 hr prior to infection. Cells were primed with 100 ng/mL Pam3CSK4 for 16 hr. Cells were then infected with PBS (Mock) or WT *S.* Typhimurium at an MOI = 20. (**A**) Release of IL-1β into the supernatant was measured by ELISA at 6 hpi. (**B**) Cell death (% cytotoxicity) was measured by lactate dehydrogenase (LDH) release assay and normalized to mock-

*Figure 4 continued on next page*

*Figure 4 continued*

infected cells at 6 hpi. (**C, D, E**) Cells were lysed at 1 hpi and 6 hpi, and bacteria were subsequently plated to calculate colony-forming units (CFUs). (**C**) CFU/well at 1 hpi. (**D**) CFU/well at 6 hpi. (**E**) Fold-change in CFU/well was calculated. Bars represent the mean for each condition. Error bars represent the standard deviation of triplicate wells from one experiment. ns – not significant, *p<0.05, **p<0.01 by unpaired t-test. Data shown are representative of at least three independent experiments.

The online version of this article includes the following source data and figure supplement(s) for figure 4:

**Source data 1.** The numerical source data that corresponds to *Figure 4*.

**Figure supplement 1.** Cytoprotection by glycine promotes *Salmonella* replication within human macrophages.

**Figure supplement 1—source data 1.** The numerical source data that corresponds to *Figure 4—figure supplement 1*.

(*Figure 5B–C*). Therefore, these results suggest that the SPI-1 T3SS contributes to *Salmonella*'s cytosolic access in human macrophages.

Since CFU assays are population-based, we next used single-cell microscopy-based methods to interrogate the subcellular localization of WT *Salmonella* in human macrophages. We employed a strain of WT *Salmonella* that constitutively expresses mCherry and maintains a reporter plasmid, pNF101, that expresses *gfp-ova* under the control of a promoter responsive to the host cytosolic metabolite glucose-6-phosphate (*Lau et al., 2019*; *Spinnenhirn et al., 2014*). We scored the number of GFP-positive/mCherry-positive bacteria (cytosolic) and GFP-negative/mCherry-positive bacteria (vacuolar) in WT, *NAIP*[-/-] and *CASP1*[-/-] THP-1 macrophages at 8 hpi by microscopy. We observed low numbers of both vacuolar and cytosolic populations of *Salmonella* in WT THP-1 cells (*Figure 5D–E*). In contrast, *NAIP*[-/-] and *CASP1*[-/-] THP-1 cells maintained significantly larger numbers of vacuolar *Salmonella*, with a substantial increase in the numbers of cytosolic *Salmonella* (*Figure 5D–E*). Taken together, these results suggest that inflammasome responses primarily restrict *Salmonella* replication within the host cell cytosol and also control bacterial replication within SCVs in human macrophages.

We next asked whether inflammasome responses also restrict the replication of *Salmonella* within the cytosol and SCV in primary hMDMs. We pretreated hMDMs with either ZVAD, to inhibit inflammasome responses mediated by caspase activity, or the vehicle control DMSO, and then infected the cells with WT *Salmonella* constitutively expressing mCherry and harboring pNF101. We observed that hMDMs treated with ZVAD contained significantly more cytosolic *Salmonella* than hMDMs treated with DMSO at 8 hpi (*Figure 5—figure supplement 1A–B*). We observed no significant differences between the vacuolar burdens of *Salmonella* in hMDMs pretreated with ZVAD or DMSO (*Figure 5—figure supplement 1A–B*). Thus, inflammasome responses appear to primarily control *Salmonella* replication within the cytosol of primary human macrophages.

## Inflammasome activation modulates the cytosolic exposure of *Salmonella* in human macrophages

Finally, to characterize the effect of inflammasomes on the subcellular niches of *Salmonella* in human macrophages at higher resolution, we used transmission electron microscopy (TEM). In WT THP-1 cells, TEM analysis revealed that the majority of bacteria resided in a membrane-bound compartment (vacuolar, white arrows) (*Figure 6A*; *Figure 6—figure supplement 1B*). In *CASP1*[-/-] THP-1 cells, we observed a more mixed population of vacuolar bacteria and bacteria that were exposed to the host cell cytosol to varying extents (*Figure 6B*; *Figure 6—figure supplement 1B*). We observed *Salmonella* free-living in the cytosol without a vacuolar membrane (fully cytosolic, black asterisk), as well as *Salmonella* exposed to the cytosol within discontinuous vacuolar membranes (partially cytosolic, cyan arrows) (*Figure 6B*; *Figure 6—figure supplement 1B*). We observed a greater proportion of cytosol-exposed *Salmonella* in *CASP1*[-/-] cells compared to WT cells (*Figure 6—figure supplement 1A*). Strikingly, these TEM results revealed distinct intracellular populations of *Salmonella* in human macrophages, providing further evidence that inflammasome signaling controls the replicative niches that *Salmonella* occupies in human macrophages.

We next sought to further characterize the cytosolic exposure of *Salmonella* in *CASP1*[-/-] THP-1 cells using electron tomography (ET), a technique that can reveal three-dimensional (3D) details about SCV membranes. Indeed, we confirmed the presence of *Salmonella* in SCVs with varying degrees of vacuolar membrane discontinuities, yielding various extents of cytosolic exposure (*Figure 6C*; *Figure 6—figure supplement 1C*; *Video 1*). These discontinuities were often numerous and found in clusters.

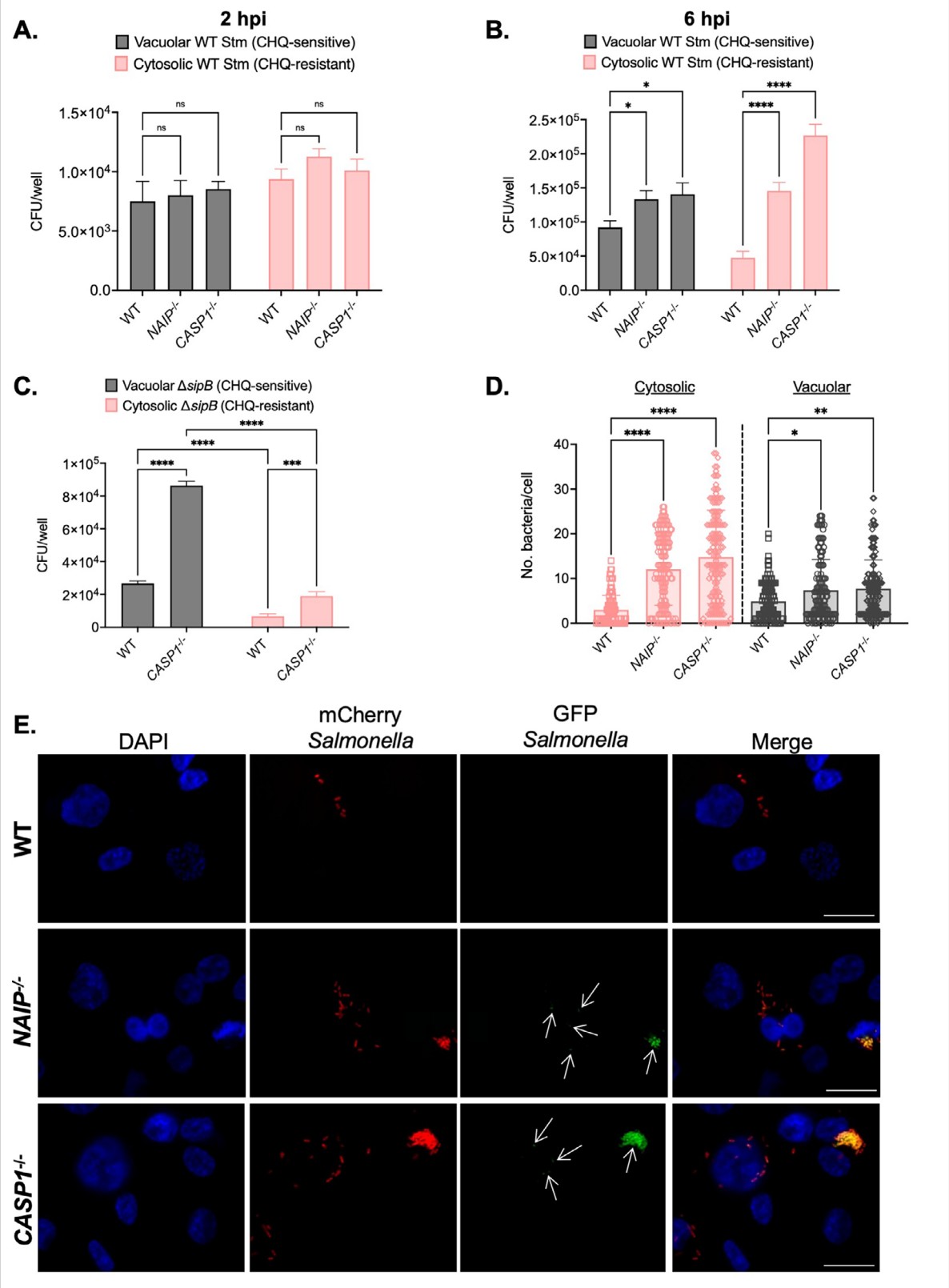

**Figure 5.** Inflammasome activation primarily controls cytosolic *Salmonella* replication in human macrophages. Wild-type (WT), *NAIP⁻/⁻* (**A, B**), and *CASP1⁻/⁻* (**A, B, C**) THP-1 monocyte-derived macrophages were primed with 100 ng/mL Pam3CSK4 for 16 hr. (**A**) Cells were then infected WT *S*. Typhimurium (**A**) at an MOI = 20. At 1 hpi, cells were left untreated or treated with 500 µM chloroquine (CHQ) for 1 hr. Then, at 2 hpi, cells were lysed, and bacteria were subsequently plated to calculate colony-forming units (CFUs). CFU/well of vacuolar (CHQ-sensitive) and cytosolic (CHQ-resistant) bacteria were

*Figure 5 continued on next page*

*Figure 5 continued*

calculated from the total bacteria. (**B, C**) Cells were then infected with WT *S.* Typhimurium (**B**) or Δ*sipB S.* Typhimurium (**C**) at an MOI = 20. At 5 hpi, cells were left untreated or treated with 500 µM CHQ for 1 hr. Then, at 6 hpi, cells were lysed, and bacteria were subsequently plated to calculate CFUs. CFU/well of vacuolar (CHQ-sensitive) and cytosolic (CHQ-resistant) bacteria were calculated from the total bacteria. (**D,E**) WT, *NAIP*[-/-], and *CASP1*[-/-] THP-1 monocyte-derived macrophages were primed with 100 ng/mL Pam3CSK4 for 16 hr. Cells were then infected with PBS (Mock) or WT *S.* Typhimurium constitutively expressing mCherry and harboring the GFP cytosolic reporter plasmid, pNF101, at an MOI = 20. Cells were fixed at 8 hpi and stained for DAPI to label DNA (blue). (**D**) The number of GFP-positive, mCherry-positive bacteria (cytosolic) per cell and the number of GFP-negative, mCherry-positive bacteria (vacuolar) per cell were scored by fluorescence microscopy. Each dot represents one infected cell. 150 total infected cells were scored for each genotype. (**E**) Representative images from 8 hpi are shown. Scale bar represents 10 µm. White arrows indicate cytosolic bacteria (GFP-positive, mCherry-positive). Bars represent the mean for each genotype (**A, B**). Error bars represent the standard deviation of triplicate wells from one experiment (**A, B**). ns – not significant, *p<0.05, ****p<0.0001 by Dunnett's multiple comparisons test (**A**) or by Tukey's multiple comparisons test (**B**). Data shown are representative of at least three independent experiments (**A, B, C**).

The online version of this article includes the following source data and figure supplement(s) for figure 5:

**Source data 1.** The numerical source data that corresponds to *Figure 5*.

**Figure supplement 1.** Inflammasome activation primarily controls the cytosolic population of *Salmonella* in primary human macrophages.

**Figure supplement 1—source data 1.** The numerical source data that corresponds to *Figure 5—figure supplement 1*.

Altogether, our TEM and ET data reveal the full spectrum of cytosolic exposure for *Salmonella* in *CASP1*[-/-] cells as a result of vacuolar membrane discontinuities. Overall, our findings indicate that inflammasome responses modulate the subcellular populations of *Salmonella*, thereby controlling the number of *Salmonella* able to replicate within the cytosol and SCV in human macrophages.

## Discussion

Our data reveal that inflammatory caspases and downstream cell lysis mediators are required to restrict *Salmonella* replication in human macrophages. Furthermore, our findings indicate that inflammasomes restrict *Salmonella* hyperreplication within the cytosol of human macrophages. Caspase-1 is required for inflammasome responses and control of intracellular *Salmonella* replication early during infection. In contrast, caspase-4 contributed minimally early during infection and instead played a larger role in inflammasome responses and restriction of *Salmonella* replication at later timepoints. We also found that GSDMD and NINJ1 were required for cell death, IL-1 cytokine release, and control of *Salmonella* replication. Finally, in the absence of these inflammasome components and effectors in human macrophages, we observed a hyperreplicating population of *Salmonella* within the cytosol, as well as increased bacterial loads within the SCV.

There are multiple downstream consequences of inflammasome activation, including pyroptosis, cytokine secretion, phagolysosomal fusion, the formation of pore-induced intracellular traps (PITs), and the generation of reactive oxygen species (ROS), which could be singly or collectively responsible for the control of intracellular *Salmonella* replication in human macrophages. Previous studies reported that inflammasome activation and inflammatory caspases limit intracellular replication of WT *Salmonella* in human epithelial cells and murine macrophages through ROS, actin polymerization, and host cell death (*Aachoui et al., 2013*; *Holly et al., 2020*; *Knodler et al., 2014a*; *Man et al., 2014a*). Interestingly, in murine macrophages, the restriction of a mutant strain of *Salmonella* that frequently enters the cytosol is dependent on caspase-1/11 but independent of IL-1 signaling and host cell death (*Thurston et al., 2016*). Moreover, caspase-1/11-dependent production of mitochondrial ROS and hydrogen peroxide contributes to the control of WT *Salmonella* replication in SCVs in murine macrophages (*Man et al., 2014a*). Future studies investigating the downstream consequences of inflammasome activation will help elucidate how inflammatory caspases and cell lysis mediators control intracellular *Salmonella* replication in human macrophages.

Whereas caspase-1 is the primary caspase that mediates inflammasome responses and control of *Salmonella* burdens in human macrophages, our data indicate that caspase-4 also plays a role at a later stage of infection. Whether caspase-4 is activated by vacuolar *Salmonella* or hyperreplicating cytosolic *Salmonella* remains an open question. Other host immune factors may also contribute to caspase-4-dependent inflammasome responses to *Salmonella*. For example, guanylate binding proteins (GBPs) can modulate inflammasome responses to intracellular LPS and impact intracellular

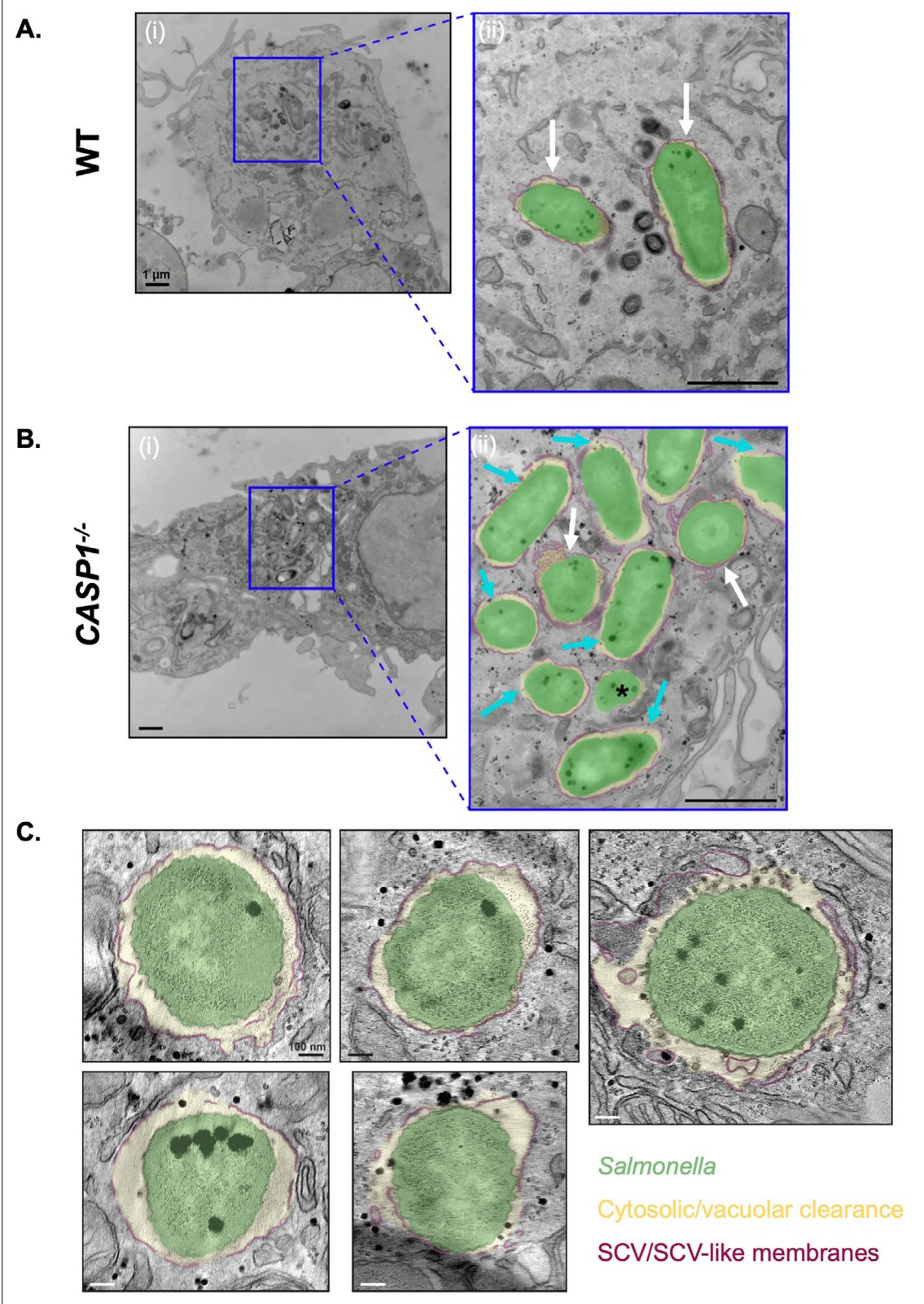

**Figure 6.** Inflammasome activation modulates the cytosolic exposure of *Salmonella* in human macrophages. (**A, B**) Wild-type (WT) and *CASP1*⁻/⁻ THP-1 monocyte-derived macrophages were primed with 100 ng/mL Pam3CSK4 for 16 hr. Cells were then infected with WT *S.* Typhimurium at an MOI = 20. At 8 hpi, cells were fixed and collected to be processed for transmission electron microscopy. Representative transmission electron micrographs are shown. *Salmonella* (green) in an *Salmonella*-containing vacuole (SCV) (maroon) with cytosolic or vacuolar clearance (yellow). Scale bar represents 1 µm [A(i), B(i)]

*Figure 6 continued on next page*

*Figure 6 continued*

or 400 nm [A(ii), B(ii)]. (**A**) WT THP-1s, (ii) is an inset from (**i**). White arrows indicate vacuolar bacteria. (**B**) *CASP1^-/-* THP-1s, (ii) is an inset from (**i**). White arrows indicate vacuolar bacteria, cyan arrows indicate partially cytosolic bacteria, and black asterisk indicates a fully cytosolic bacterium. (**C**) *CASP1^-/-* THP-1 monocyte-derived macrophages were primed with 100 ng/mL Pam3CSK4 for 16 hr. Cells were then infected with WT *S.* Typhimurium at an MOI = 20. At 8 hpi, cells were fixed and collected to be processed for transmission electron microscopy. Representative tomogram slices are shown, depicting *Salmonella* (green) in an SCV with a discontinuous vacuolar membrane (maroon).

The online version of this article includes the following source data and figure supplement(s) for figure 6:

**Figure supplement 1.** Characterization of *Salmonella* subpopulations in human macrophages.

**Figure supplement 1—source data 1.** The numerical source data that corresponds to *Figure 6—figure supplement 1*.

bacterial replication (*Degrandi et al., 2007*; *Fisch et al., 2020*; *Kim et al., 2011*; *Kutsch et al., 2020*; *Meunier et al., 2014*; *Pilla et al., 2014*; *Santos et al., 2020*; *Tietzel et al., 2009*; *Wandel et al., 2020*).

The impact of GSDMD on the restriction of intracellular *Salmonella* replication may be due to downstream responses, such as IL-1 release and pyroptosis. Moreover, pyroptosis can induce PIT formation, which traps and damages intracellular bacteria, rendering them more susceptible to immune defenses, such as neutrophil-mediated killing (*Jorgensen et al., 2016*). It is possible that pyroptosis leads to the development of PITs in human macrophages during *Salmonella* infection, thereby limiting *Salmonella*'s intracellular replication. In addition, the cleaved N-terminal fragment of GSDMD can directly kill bacteria (*Ding et al., 2016*; *Liu et al., 2016*; *Wang et al., 2019*), raising the possibility that GSDMD directly targets cytosolic *Salmonella* to mediate restriction. A recent study revealed that GSDMD binds and permeabilizes mitochondrial membranes to mediate pyroptosis (*Miao et al., 2023*). Still, it remains unknown whether GSDMD binds or permeabilizes the SCV, thereby affecting vacuolar integrity and influencing intracellular *Salmonella* replication.

Our data indicate that inflammasome responses primarily control *Salmonella* replication within the cytosol of human macrophages and also within SCVs. How inflammasome activation inhibits *Salmonella* replication in the cytosol of human macrophages remains unknown. *Salmonella* hyperreplicates in the cytosol of human epithelial cells, which are unable to undergo caspase-1-dependent inflammasome responses (*Holly et al., 2020*; *Knodler et al., 2014a*; *Knodler et al., 2010*; *Knodler et al., 2014b*; *Naseer et al., 2022b*). Our study suggests that *CASP1^-/-* THP-1 macrophages behave similarly to human IECs, as they support *Salmonella* hyperreplication in the cytosol in the absence of caspase-1. Inflammasome activation could curtail cytosolic replication of *Salmonella* through host cell death, direct GSDMD targeting of cytosolic *Salmonella,* and/or another effector mechanism. Alternatively, inflammasome activation could regulate the cytosolic access of *Salmonella* through host factors that directly damage the SCV, such as pore-forming proteins like GSDMD.

The bacterial factors that facilitate *Salmonella*'s cytosolic lifestyle in human macrophages remain largely unknown. Our findings indicate that the SPI-1 T3SS is partially required for *Salmonella* to access the cytosol in THP-1 macrophages.

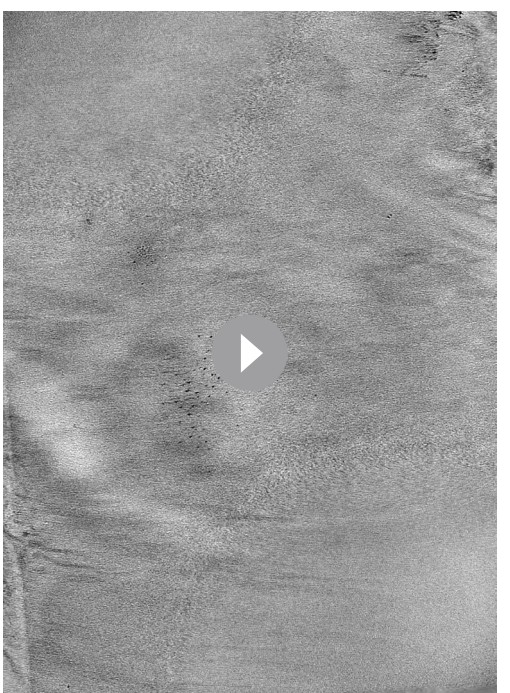

**Video 1.** Characterization of the cytosolic exposure of *Salmonella* in *CASP1^-/-* human macrophages. *CASP1^-/-* THP-1 monocyte-derived macrophages were primed with 100 ng/mL Pam3CSK4 for 16 hr. Cells were then infected with wild-type (WT) *S.* Typhimurium at an MOI = 20. At 8 hpi, cells were fixed and collected to be processed for transmission electron microscopy. Representative tomogram shown, depicting *Salmonella* in an *Salmonella*-containing vacuole (SCV) with a discontinuous vacuolar membrane.
https://elifesciences.org/articles/90107/figures#video1

In epithelial cells, the SPI-1 T3SS SopE, SopB, and SipA enable *Salmonella* to escape the SCV and efficiently colonize and replicate in the cytosol (*Chong et al., 2019*; *Klein et al., 2017a*; *Röder and Hensel, 2020*). Future studies will elucidate whether these same effectors also support *Salmonella*'s escape into the cytosol and cytosolic replication in human macrophages.

Our data also indicate that there is considerable single-cell heterogeneity in terms of total intracellular bacterial burdens as well as proportions of vacuolar and cytosolic *Salmonella* in human macrophages using single-cell microscopic analysis. Several factors could account for this phenotypic heterogeneity, including expression levels of inflammasome components, the amount of translocated *Salmonella* ligands, or the extent of inflammasome activation, all of which might differ in individual cells and are masked in bulk-population assays. Notably, heterogeneity in *Salmonella* gene expression, intracellular *Salmonella* populations, and intracellular *Salmonella* proliferation and viability has been previously observed in human epithelial cells and murine macrophages (*Helaine et al., 2010*; *Helaine et al., 2014*; *Knodler et al., 2014b*; *Malik Kale et al., 2012*; *Powers et al., 2021*).

In this study, we utilized tissue culture models to examine intracellular *Salmonella* replication in human macrophages. These in vitro systems allow for precise control of experimental conditions and, therefore, serve as powerful tools to interrogate the molecular mechanisms underlying inflammasome responses and *Salmonella* replication in both immortalized and primary human cells. Still, there are limitations of tissue culture models, as they lack the inherent complexity of tissues and organs in vivo. To assess whether our findings reflect *Salmonella* dynamics in the mammalian host, it will be important to complement our studies and extend the implications of our work using approaches that model more complex systems, such as organoids or organ explant models co-cultured with immune cells, and in vivo techniques, such as humanized mouse models.

Collectively, our results reveal that inflammasome responses restrict intracellular *Salmonella* replication, particularly within the cytosol of human macrophages. These findings are in contrast to mouse macrophages, where inflammasomes are activated but only marginally restrict the intracellular replication of WT *Salmonella*. Our findings provide insight into how human macrophages leverage inflammasomes to restrict *Salmonella* intracellular replication. Moreover, our work offers a basis for future studies to investigate how inflammasome activation modulates the subcellular localization of bacterial replicative niches within host cells.

## Materials and methods

### Ethics statement

All experiments on primary hMDMs were performed in compliance with the requirements of the US Department of Health and Human Services and the principles expressed in the Declaration of Helsinki. hMDMs were derived from samples obtained from the University of Pennsylvania Human Immunology Core, and they are considered to be a secondary use of deidentified human specimens and are exempt via Title 55 Part 46, Subpart A of 46.101 (b) of the Code of Federal Regulations.

### Cell culture of THP-1 cells

THP-1 cells (TIB-202; American Type Culture Collection) were maintained in RPMI supplemented with 10% (vol/vol) heat-inactivated FBS, 0.05 nM β-mercaptoethanol, 100 IU/mL penicillin, and 100 µg/mL streptomycin at 37°C in a humidified incubator. Two days prior to experimentation, the cells were replated in media without antibiotics in a 48-well plate at a concentration of $2 \times 10^5$ cells/well or in a 24-well plate at a concentration of $3.5 \times 10^5$ cells per well and incubated with phorbol 12-myristate 13-acetate (PMA) for 24 hr to allow differentiation into macrophages. Then, macrophages were either left unprimed or primed with 100 ng/mL Pam3CSK4 (Invivogen) for 16–20 hr prior to bacterial infections. For fluorescence microscopy experiments, cells were plated on glass coverslips in a 24-well plate.

### Cell culture of primary hMDMs

Purified human monocytes from deidentified healthy human donors were obtained from the University of Pennsylvania Human Immunology Core. The monocytes were cultured in RPMI supplemented with 10% (vol/vol) heat-inactivated FBS, 2 mM L-glutamine, 100 IU/mL penicillin, 100 µg/ml streptomycin, and 50 ng/ml recombinant human M-CSF (Gemini Bio-Products). Cells were cultured for 4 d

in 10 mL of media in at a concentration of 4–5 × 10$^5$ cells/mL in 10 cm-dishes. Then, 10 mL of fresh growth media was added for an additional 2 d to promote complete differentiation into macrophages. One day prior to infection, the cells were rinsed with cold PBS, gently detached with trypsin-EDTA (0.05%), and replated in media without antibiotics and with 25 ng/mL human M-CSF in a 48-well plate at a concentration of 1 × 10$^5$ cells per well or in a 24-well plate at a concentration of 2 × 10$^5$ cells per well. For fluorescence microscopy, cells were replated on glass coverslips in a 24-well plate. For experiments involving LPS, cells were primed with 500 ng/mL LPS (Sigma-Aldrich) for 3 hr prior to bacterial infections.

### Inhibitor experiments

Human macrophages were treated 1 hr prior to infection at the indicated concentrations with the following inhibitors: 20 μM of the pan-caspase inhibitor Z-VAD(OMe)-FMK (SM Biochemicals; SMFMK001), 25 μM of the caspase-1 inhibitor Ac-YVAD-cmk (Sigma-Aldrich; SML0429), and 40 μM of the GSDMD inhibitor disulfiram (Sigma). DMSO treatment was used as a vehicle control with these inhibitors. To prevent cell lysis, cells were treated with 20 mM glycine (Fisher Scientific) for 30 min prior to infection. Distilled water was used as a vehicle control.

### Bacterial strains and growth conditions

*Salmonella enterica* serovar Typhimurium SL1344 WT was routinely grown shaking overnight at 37°C in Luria-Bertani (LB) broth with streptomycin (100 μg/mL). For infection of cultured cells, the overnight culture was diluted in LB with streptomycin (100 μg/mL) containing 300 mM NaCl and grown standing for 3 hr at 37°C to induce SPI-1 expression (*Lee and Falkow, 1990*), unless otherwise noted, when the overnight culture of *Salmonella* was grown to stationary phase and subsequently used to infect cultured cells.

SL1344 WT *glmS::Ptrc-mCherryST::FRT* pNF101 *Lau et al., 2019* was kindly provided by Dr. Leigh Knodler. This strain constitutively expresses *mCherry* that is chromosomally encoded. It also harbors the *PuhpT-gfpova* plasmid, pNF101, where expression of GFP is under the control of the glucose-6-phosphate responsive *uhpT* promoter derived from *Shigella flexneri*. SL1344 WT *glmS::Ptrc-mCherryST::FRT* pNF101 was grown shaking overnight at 37°C in Luria-Bertani (LB) broth with streptomycin (100 μg/mL) and ampicillin (100 μg/mL). For SPI-1 induction prior to infection, the overnight culture was diluted in LB with streptomycin (100 μg/mL) and ampicillin (100 μg/mL) that also contained 300 mM NaCl and then grown standing for 3 hr at 37°C (*Lee and Falkow, 1990*).

### Bacterial infections

Overnight cultures of *Salmonella* were diluted into LB broth with streptomycin (100 μg/mL) containing 300 mM NaCl and grown for 3 hr standing at 37°C to induce SPI-1 gene expression (*Lee and Falkow, 1990*). Bacterial cultures were then pelleted at 6010 × g for 3 min, washed once with PBS, and resuspended in PBS. Human macrophages were infected with *Salmonella* at a multiplicity of infection (MOI) of 20. Infected cells were centrifuged at 290 × g for 10 min and incubated at 37°C. At 30 min post-infection, the cells were treated with 100 μg/ml of gentamicin to kill any extracellular *Salmonella*, unless otherwise noted when the cells were treated with 25 μg/ml of gentamicin. Then, the infection proceeded at 37°C for 6–8 hr, as indicated. For all experiments, control cells were mock-infected with PBS.

### Bacterial intracellular burden assay

Cells were infected with WT *Salmonella* as described above at an MOI of 20. Then, 30 min post-infection, cells were treated with 100 μg/ml of gentamicin to kill any extracellular bacteria. 1 hr post-infection, the media was replaced with fresh media containing 10 μg/ml of gentamicin or no gentamicin, as indicated. At the indicated time points, the infected cells were lysed with PBS containing 0.5% Triton to collect all intracellular *Salmonella*. Harvested bacteria were serially diluted in PBS and plated on LB agar plates containing streptomycin (100 μg/ml) to enumerate colony-forming units (CFUs). Plates were incubated overnight at 37°C and CFUs were subsequently counted.

## Chloroquine (CHQ) resistance assay

THP-1 macrophages were infected in 48-well plates as described above. For each timepoint, triplicate wells were incubated in the presence of CHQ (500 µM) and gentamicin (100 ng/mL) for 1 hr to quantify the CHQ-resistant bacteria (Bârzu et al., 1997; Fernandez et al., 2001; Klein et al., 2017b; Knodler et al., 2014b; Zychlinsky et al., 1994). Another triplicate wells were incubated with gentamicin (100 ng/mL) only to quantify the total intracellular bacteria. At the indicated time points, the infected cells were lysed with PBS containing 0.5% Triton to collect intracellular *Salmonella*. Harvested bacteria were serially diluted in PBS and plated on LB agar plates containing streptomycin (100 µg/ml) to enumerate CFUs. Plates were incubated overnight at 37°C and CFUs were subsequently counted.

## ELISAs

Harvested supernatants from infected human macrophages were assayed for cytokine levels using ELISA kits for human IL-1α (R&D Systems), IL-18 (R&D Systems), IL-1β (BD Biosciences), and TNF-α (R&D Systems).

## LDH cytotoxicity assays

Harvested supernatants from infected human macrophages were assayed for cytotoxicity by quantifying the loss of cellular membrane integrity via lactate dehydrogenase (LDH) activity. LDH release was measured using an LDH Cytotoxicity Detection Kit (Clontech) according to the manufacturer's instructions and normalized to mock-infected cells.

## siRNA-mediated knockdown of genes

All Silencer Select siRNA oligos were purchased from Ambion (Life Technologies). Individual siRNA targeting *NINJ1* (ID# s9556) was used. The two Silencer Select negative control siRNAs (Silencer Select Negative Control No. 1 siRNA and Silencer Select Negative Control No. 2 siRNA) were used as a control. Three days prior to the infection, 30 nM of siRNA was transfected into the human macrophages using Lipofectamine RNAiMAX transfection reagent (Thermo Fisher Scientific) following the manufacturer's protocol. Then, 24 hr after transfection, the media was replaced with fresh media containing antibiotics. Finally, 16 hr before infection, the media was replaced with fresh antibiotic-free media containing 100 ng/ml Pam3CSK4.

## Quantitative RT-PCR analysis

RNA was isolated using the RNeasy Plus Mini Kit (Qiagen) following the manufacturer's protocol. Human macrophages were lysed in 350 µL RLT buffer with β-mercaptoethanol and centrifuged through a QIAshredder spin column (Qiagen). cDNA was synthesized from isolated RNA using SuperScript II Reverse Transcriptase (Invitrogen) following the manufacturer's protocol. Quantitative PCR was conducted with the CFX96 real-time system from Bio-Rad using the SsoFast EvaGreen Supermix with Low ROX (Bio-Rad). To calculate knockdown efficiency, mRNA levels of siRNA-treated cells were normalized to the housekeeping gene *HPRT* and control siRNA-treated cells using the $2^{-\Delta\Delta CT}$ (cycle threshold) method (Livak and Schmittgen, 2001). The following primers from PrimerBank were used. The PrimerBank identifications are *NINJ1* (148922910c1), *CASP4* (73622124c2) and *HPRT* (164518913c1); all primers listed as 5′–3′:

*NINJ1* forward: TCAAGTACGACCTTAACAACCCG
*NINJ1* reverse: TGAAGATGTTGACTACCACGATG
*CASP4* forward: TCTGCGGAACTGTGCATGATG
*CASP4* reverse: TGTGTGATGAAGATAGAGCCCAT
*HPRT* forward: CCTGGCGTCGTGATTAGTGAT
*HPRT* reverse: AGACGTTCAGTCCTGTCCATAA

## Immunoblot analysis

Cell lysates were harvested for immunoblot analysis by adding 1X SDS/PAGE sample buffer to cells. All protein samples (lysates) were boiled for 5 min. Samples were separated by SDS/PAGE on a 12% (vol/vol) acrylamide gel and transferred to PVDF Immobilon-P membranes (Millipore). Primary antibodies specific for caspase-4 (4450S; Cell Signaling) and β-actin (4967L; Cell Signaling) and HRP-conjugated

secondary antibodies anti-mouse IgG (F00011; Cell Signaling) and anti-rabbit IgG (7074S; Cell Signaling) were used. ECL Western Blotting Substrate (Pierce Thermo Scientific) was used as the HRP substrate for detection.

## Fluorescent microscopy of intracellular *Salmonella*

Primary hMDMs or THP-1 cells were plated on glass coverslips in a 24-well plate as described above. Cells were either infected with WT *Salmonella* constitutively expressing GFP (Sl1344 harboring pFPV25.1) *Valdivia and Falkow, 1996*, or WT *Salmonella* constitutively expressing mCherry with a cytosolic GFP reporter (SL1344 *glmS::Ptrc-mCherryST::FRT* pNF101) *Lau et al., 2019* at an MOI of 20 as described above. At the indicated timepoints following infection, cells were washed 2 times with PBS and then fixed with 4% paraformaldehyde for 10 min. Following fixation, cells were mounted on glass slides with DAPI mounting medium (Sigma Fluoroshield). Coverslips were imaged on an inverted fluorescence microscope (IX81; Olympus), and the images were collected using a high-resolution charge-coupled-device camera (FAST1394; QImaging) at a magnification of 100×. All images were analyzed and presented using SlideBook (version 5.0) software (Intelligent Imaging Innovations, Inc) and ImageJ software. For experiments with WT *Salmonella* constitutively expressing GFP, the proportion of infected cells containing GFP-expressing *Salmonella* (green) were scored by counting 50 infected cells per coverslip. 150 total infected cells were scored for each condition. For experiments with WT *Salmonella* constitutively expressing mCherry with a cytosolic GFP reporter, the proportion of infected cells containing GFP-positive *Salmonella* (cytosolic) and GFP-negative, mCherry-positive *Salmonella* (vacuolar) were scored by counting 50 infected cells per coverslip. 150 total infected cells were scored for each condition.

## Transmission electron microscopy

Two days prior to experimentation, $2 \times 10^6$ cells THP-1 cells were replated in 10-cm dishes in media without antibiotics and incubated with phorbol 12-myristate 13-acetate (PMA) for 24 hr to allow differentiation into macrophages. Then, macrophages were primed with 100 ng/mL Pam3CSK4 (Invivogen) for 16 hr prior to bacterial infection. Cells were infected with WT *Salmonella* at an MOI of 20 as described above. At 8 hpi, the media was aspirated, and the cells were fixed with 2.5% glutaraldehyde, 2.0% paraformaldehyde in 0.1M sodium cacodylate buffer, pH 7.4. Then, at the Electron Microscopy Resource Laboratory in the Perelman School of Medicine, after subsequent buffer washes, the samples were post-fixed in 2.0% osmium tetroxide with 1.5% $K_3Fe(CN)_6$ for 1 hr at room temperature, and rinsed in $dH_2O$. After dehydration through a graded ethanol series, the tissue was infiltrated and embedded in EMbed-812 (Electron Microscopy Sciences, Fort Washington, PA). Thin sections were stained with uranyl acetate and SATO lead and examined with a JEOL 1010 electron microscope fitted with a Hamamatsu digital camera and AMT Advantage NanoSprint500 software.

## Electron tomography

Electron tomography was performed at room temperature on a Thermo Fisher Krios G3i TEM equipped with a 300 keV field emission gun. Imaging was performed using the SerialEM software (*Mastronarde, 2005*) on a K3 direct electron detector (*Xuong et al., 2007*) (Gatan Inc, Pleasanton, CA, USA) operated in the electron-counted mode. We additionally used the Gatan Imaging Filter (Gatan Inc, Pleasanton, CA, USA) with a slit width of 20 eV to increase contrast by removing inelastically scattered electrons (*Krivanek et al., 1995*). After initially assessing the infected macrophages at lower magnifications, tilt series were collected at a magnification of 26,000X (with a corresponding pixel size of 3.38 Å) and a defocus of −6 μm. Precooking the target areas with a total dosage of 1000–1500 e/Å² was required to minimize sample shrinking and drifting during tilt series collection. A bi-directional tilt scheme was employed with 2° increments and 120° span (−60° to +60°). The cumulative dose of each tilt-series was in the order of ~500 e−/Å2. Once acquired, tilt series were aligned (using the patch tracking function) and reconstructed into tomograms, both using the IMOD software package (*Kremer et al., 1996*). After careful assessment of the three-dimensional (3-D) tomograms, optimal 2-D slices were chosen for figure presentations. Color overlays provide additional assistance with interpretation.

## Statistical analysis

Prism 9.5.0 (GraphPad Software) was utilized for the graphing of data and all statistical analyses. Statistical significance for experiments with THP-1 macrophages and hMDMs was determined using the appropriate test and are indicated in each figure legend. Differences were considered statistically significant if the p-value was <0.05.

## Acknowledgements

We thank members of Igor Brodsky's and Sunny Shin's laboratories for their scientific discussions and continuous support. We thank Leigh Knodler for providing SL1344 *glmS::Ptrc-mCherryST::FRT* pNF101 and Cornelius Taabazuing for providing *CASP1*[-/-] and *GSDMD*[-/-] THP-1 cells. We thank the Human Immunology Core of the Penn Center for AIDS Research and Abramson Cancer Center for providing purified primary human monocytes. We also thank the Electron Microscopy Resource Lab (EMRL) at the Perelman School of Medicine, University of Pennsylvania for TEM specimen processing, sectioning, and staining along with microscopy training. Additionally, we thank the EMRL for allowing us access and usage of the transmission electron microscope, JEOL JEM-1010. Furthermore, we thank Gordon Ruthel at the Penn Vet Imaging Core and Ronit Schwartz for providing helpful insights on fluorescence microscopy. Lastly, we thank Marcia Goldberg and Mateusz Szczerba at Harvard Medical School for the helpful protocols and advice on the modifications of gentamicin use in the bacterial intracellular burden assay. This work was supported by National Institutes of Health (NIH)/National Institute of Allergy and Infectious Diseases (NIAID) grants: AI151476, AI118861, and AI123243 (SS), AI128630, AI163596, and AI139102 (IEB). This work was also supported by the Burroughs-Wellcome Fund Investigators in the Pathogenesis of Infectious Disease Award (SS and IEB), the National Science Foundation Graduate Fellowships DGE-1321851 (MSE) and DGE-1845298 (ARB), the NIH/NIGMS grant T32GM07229 (EAO), the David and Lucile Packard Fellowship for Science and Engineering 2019–69,645 (Y-WC), and the Pennsylvania Department of Health FY19 Health Research Formula Fund (Y-WC). The funders had no role in study design, data collection and analysis, decision to publish, or preparation of the manuscript.

## Additional information

### Funding

| Funder | Grant reference number | Author |
|---|---|---|
| National Science Foundation Graduate Research Fellowship Program | DGE- 1321851 | Marisa S Egan |
| National Institute of Allergy and Infectious Diseases | AI151476 | Sunny Shin |
| National Institute of Allergy and Infectious Diseases | AI118861 | Sunny Shin |
| National Institute of Allergy and Infectious Diseases | AI123243 | Sunny Shin |
| National Institute of Allergy and Infectious Diseases | AI128630 | Igor E Brodsky |
| National Institute of Allergy and Infectious Diseases | AI163596 | Igor E Brodsky |
| National Institute of Allergy and Infectious Diseases | AI139102 | Igor E Brodsky |
| Burroughs Wellcome Fund | PATH Award | Yi-Wei Chang |

| Funder | Grant reference number | Author |
|---|---|---|
| National Science Foundation Graduate Research Fellowship Program | DGE-1845298 | Antonia R Bass |
| National Institute of General Medical Sciences | T32GM07229 | Emily A O'Rourke |
| David and Lucile Packard Foundation | 2019-69645 | Yi-Wei Chang |
| Pennsylvania Department of Health | FY19 Health Research Formula Fund | Yi-Wei Chang |
| American Heart Association | 10.58275/aha.24pre1199554.pc.gr.190717 | Emily A O'Rourke |

The funders had no role in study design, data collection and interpretation, or the decision to submit the work for publication.

## Author contributions

Marisa S Egan, Conceptualization, Resources, Data curation, Software, Formal analysis, Funding acquisition, Validation, Investigation, Visualization, Methodology, Writing – original draft, Writing – review and editing; Emily A O'Rourke, Data curation, Formal analysis, Funding acquisition, Validation, Investigation, Visualization, Writing – original draft, Writing – review and editing; Shrawan Kumar Mageswaran, Resources, Data curation, Formal analysis, Validation, Investigation, Visualization, Methodology, Writing – original draft, Writing – review and editing; Biao Zuo, Inna Martynyuk, Methodology, Writing – review and editing; Tabitha Demissie, Emma N Hunter, Formal analysis, Validation, Investigation, Writing – review and editing; Antonia R Bass, Conceptualization, Data curation, Formal analysis, Funding acquisition, Validation, Investigation, Methodology, Writing – review and editing; Yi-Wei Chang, Igor E Brodsky, Conceptualization, Supervision, Funding acquisition, Methodology, Writing – review and editing; Sunny Shin, Conceptualization, Supervision, Funding acquisition, Project administration, Writing – review and editing

## Author ORCIDs

Marisa S Egan ⓘ https://orcid.org/0000-0001-6191-5552
Emily A O'Rourke ⓘ http://orcid.org/0000-0001-6736-7374
Biao Zuo ⓘ http://orcid.org/0009-0005-9115-7170
Emma N Hunter ⓘ http://orcid.org/0000-0001-7549-9190
Antonia R Bass ⓘ http://orcid.org/0000-0002-5623-0572
Yi-Wei Chang ⓘ http://orcid.org/0000-0003-2391-473X
Igor E Brodsky ⓘ https://orcid.org/0000-0001-7970-872X
Sunny Shin ⓘ https://orcid.org/0000-0001-5214-9577

Reviewer #1 (Public review): https://doi.org/10.7554/eLife.90107.3.sa1
Reviewer #2 (Public review): https://doi.org/10.7554/eLife.90107.3.sa2
Reviewer #3 (Public review): https://doi.org/10.7554/eLife.90107.3.sa3
Author response https://doi.org/10.7554/eLife.90107.3.sa4

# Additional files

## Supplementary files

MDAR checklist

## Data availability

All data generated or analyzed during this study are included in the manuscript, supporting files, and source data files.

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
