## [Editor Report · eLife Assessment]

This paper provides **fundamental** insights into the control of *Salmonella* within human macrophages, with **convincing** evidence that *Salmonella* can replicate in the macrophage cytosol in the absence of inflammasome signaling. This paper, which improves our understanding of how the immune system fights bacterial infections, will be of broad interest to cell biologists, immunologists and microbiologists.

---

## [Referee Report · Reviewer #1 (Public review)]

Summary:

In this excellent manuscript by Egan et al., the authors very carefully dissect the roles of inflammasome components in restricting *Salmonella* Typhimurium (STm) replication in human macrophages. They show that caspase-1 is essential to mediating inflammasome responses and that caspase-4 contributes to bacterial restriction at later time points. The authors show very clear roles for the host proteins that mediate terminal lysis, gasdermin D and ninjurin-1. The unique finding in this study is that in the absence of inflammasome responses, *Salmonella* hypereplicates within the cytosol of macrophages. These findings suggest that caspase-1 and possibly caspase-4 play roles in restricting the replication of Salmonella in the cytosol as well as in the Salmonella containing vacuole.

Strengths:

(1) The genetic and biochemical approaches have shown for the first time in human macrophages that the caspase-1-GSDMD-NINJ1 axis is very important for restricting intracellular STm replication. In addition, they demonstrate a later role for Casp4 in control of intracellular bacterial replication.

(2) In addition, they show that in macrophages deficient in the caspase-1-GSDMD-NINJ1 axis that STm are found replicating in the cytosol, which is a novel finding. The electron microscopy is convincing that STm are in the cytosol.

(3) The authors go on to use a chloroquine resistance assay to show that inflammasome signaling also restricts STm within SCVs in human macrophages.

(4) Finally, they show that the Type 3 Secretion System encoded on *Salmonella* Pathogenicity Island 1 contributes to STm's cytosolic access in human macrophages.

Weaknesses:

(1) Their results with human macrophages suggest that there are differences between murine and human macrophages in inflammasome-mediated restriction of STm growth. For example, Thurston et al. showed that in murine macrophages that inflammasome activation controls the replication of mutant STm that aberrantly invades the cytosol, but only slightly limits replication of WT STm. In contrast, here the authors found that primed human macrophages rely on caspase-1, gasdermin D and ninjurin-1 to restrict WT STm. I wonder if the priming of the human macrophages in this study could account for the differences in these studies. Along those lines, do the authors see the same results presented in this study in the absence of priming the macrophages with Pam3CSK4. I think that determining whether the control of intracellular STm replication is dependent on priming is very important. Another difference with the Thurston et al. paper is the way that the STm inoculum was prepared - stationary phase bacteria that were opsonized. Could this also account for differences between the two studies rather than differences between murine and human macrophages in inflammasome-dependent control of STm?

(2) The authors show that the pore-forming proteins GSDMD and Ninj1 contribute to control of STm replication in human macrophages. Is it possible that leakage of gentamicin from the media contributes to this control?

(3) One major question that remains to be answered is whether casp-1 plays a direct role in the intracellular localization of STm. If the authors quantify the percentage of vacuolar vs. cytosolic bacteria at early time points in WT and casp-1 KO macrophages, would that be the same in the presence and absence of casp-1? If so, then this would suggest that there is a basal level of bacterial-dependent lysis of the SCV and in WT macrophages the presence of cytosolic PAMPS trigger cell death and bacteria can't replicate in the cytosol. However, in the inflammasome KO macrophages, the host cell remains alive and bacteria can replicate in the cytosol.

Comments on revisions:

The authors have addressed my previous concerns. The addition of the statements indicating the limitations of the study are an important addition.

---

## [Referee Report · Reviewer #2 (Public review)]

Summary:

This work addresses the question of how human macrophages restrict intracellular replication of *Salmonella*.

Strengths:

Through a series of genetic knockouts and using specific inhibitors, Egan et al. demonstrated that the inflammasome components caspase-1, caspase-4, gasdermin D (GSDMD), and the final lytic death effector ninjurin-1 (NINJ1) are required for control of *Salmonella* replication in human macrophages. Interestingly, caspase-1 proved crucial in restricting *Salmonella* early during infection, whereas caspase-4 was essential in the later stages of infection. Furthermore, using a chloroquine resistance assay and state-of-the-art microscopy, the authors found that NAIP receptor and caspase-1 mostly regulate replication of cytosolic bacteria, with smaller, yet significant, impact on the vacuolar bacteria.

The finding that inflammasomes are critical in the restriction of replication of intracellular *Salmonella* in human macrophages contrasts with the published minimal role of inflammasomes in restriction of replication of intracellular *Salmonella* in murine macrophages. Some of these differences could be due to differences in the methodologies used in the two studies. However, the findings suggest yet another example of interspecies and intercellular differences in regulation of bacterial infections by the immune system.

Comments on revisions:

The authors may wish to comment that the measurements of released cytokines by ELISA do not discriminate between active and full-length forms of the cytokines.

---

## [Referee Report · Reviewer #3 (Public review)]

The manuscript by Egan and coworkers investigates how Caspase-1 and Caspase-4 mediated cell death affects replication of *Salmonella* in human THP-1 macrophages in vitro.

Overall evaluation:

Strength of the study include the use of human cells, which exhibit notable differences (e.g., Caspase 11 vs Caspase-4/5) compared to commonly used murine models. Furthermore, the study combines inhibitors with host and bacterial genetics to elucidate mechanistic links.

Comments on revisions:

The authors have addressed my comments regarding the previous submission.

---

## [Author Response]

The following is the authors’ response to the original reviews.

**Reviewer #1:**
(1) Their results with human macrophages suggest that there are differences between murine and human macrophages in inflammasome-mediated restriction of STm growth. For example, Thurston et al. showed that in murine macrophages that inflammasome activation controls the replication of mutant STm that aberrantly invades the cytosol, but only slightly limits replication of WT STm. In contrast, here the authors found that primed human macrophages rely on caspase-1, gasdermin D and ninjurin-1 to restrict WT STm. I wonder if the priming of the human macrophages in this study could account for the differences in these studies. Along those lines, do the authors see the same results presented in this study in the absence of priming the macrophages with Pam3CSK4. I think that determining whether the control of intracellular STm replication is dependent on priming is very important.

We thank the Reviewer for their careful attention to our manuscript and for their thoughtful comments. We have addressed this question about the impact of priming by repeating the bacterial intracellular burden assays in unprimed WT and CASP1-/- THP-1 cells. We have added additional figures to the manuscript to address this: Figure 1 – Figure Supplement 3. Under unprimed conditions, CASP1-/- cells still harbored significantly higher bacterial burdens at 6 hpi and a significant fold-increase in bacterial CFUs compared to WT cells. These results suggest that the caspase-1-mediated restriction of intracellular *Salmonella* replication in human macrophages is independent of priming.

(2) Another difference with the Thurston et al. paper is the way that the STm inoculum was prepared - stationary phase bacteria that were opsonized. Could this also account for differences between the two studies rather than differences between murine and human macrophages in inflammasome-dependent control of STm?

We thank the Reviewer for this excellent suggestion. To address this possibility, we repeated the bacterial intracellular burden assays in WT and CASP1-/- THP-1 cells using stationary phase bacteria. We infected WT and CASP1-/- THP-1 cells with stationary phase *Salmonella*, and we subsequently assayed for intracellular bacterial burdens. These data have now been added to the manuscript in Figure 1 – Figure Supplement 4. Interestingly, we did not observe any fold-change in the bacterial colony forming units in both the WT and CASP1-/- THP-1 cells for the stationary phase *Salmonella*. These data indicate that by 6 hours postinfection, *Salmonella* do not replicate efficiently in human macrophages unless grown under SPI-1-inducing conditions. Furthermore, these results suggest that differences in how the *Salmonella* inoculum is prepared may contribute to the discrepancies between our study and previous studies, as noted by the Reviewer.

(3) The authors show that the pore-forming proteins GSDMD and Ninj1 contribute to control of STm replication in human macrophages. Is it possible that leakage of gentamicin from the media contributes to this control?

Response: We thank the Reviewer for their insightful comment. We have addressed this question on the impact of gentamicin by repeating the bacterial intracellular burden assays using a lower concentration of gentamicin in combination with extensively washing the cells with RPMI media to remove the gentamicin. WT and CASP1-/- THP-1 cells were infected with WT *Salmonella*. Then, at 30 minutes post-infection, cells were treated with 25 μg/ml of gentamicin to kill any extracellular bacteria. At 1 hour post-infection (hpi), the cells were washed for a total of five times with fresh RPMI to remove the gentamicin, and then the media was replaced with fresh media containing no gentamicin. In parallel, we also treated cells with 100 μg/ml of gentamicin at 30 minutes post-infection, washed the cells five times with fresh RPMI at 1 hpi to remove the gentamicin, and then replaced the media with fresh media containing 10 μg/ml of gentamicin. This data has now been included in the manuscript as Figure 1 – Figure Supplement 5. We observed similar levels in the intracellular bacterial burdens at 1 hpi and 6 hpi and a fold-increase in bacterial colony forming units in CASP1-/- cells compared to WT cells across both gentamicin conditions, suggesting that gentamicin appears to not contribute to the intracellular control of *Salmonella* replication in human macrophages. Of note, we also tried repeating the bacterial intracellular burden assays without gentamicin, using only washes to remove extracellular at 1 hpi; however, under these experimental conditions, we observed high levels of extracellular *Salmonella*. Therefore, we relied on using a lower concentration of gentamicin to kill extracellular *Salmonella* in conjunction with extensive washing to remove the gentamicin for the remainder of the infection.

(4) One major question that remains to be answered is whether casp-1 plays a direct role in the intracellular localization of STm. If the authors quantify the percentage of vacuolar vs. cytosolic bacteria at early time points in WT and casp-1 KO macrophages, would that be the same in the presence and absence of casp-1? If so, then this would suggest that there is a basal level of bacterial-dependent lysis of the SCV and in WT macrophages the presence of cytosolic PAMPS trigger cell death and bacteria can't replicate in the cytosol. However, in the inflammasome KO macrophages, the host cell remains alive and bacteria can replicate in the cytosol.

We thank this Reviewer for raising this important point. We have addressed this experimentally by quantifying the percentage of vacuolar vs. cytosolic *Salmonella* at 2 hpi in WT, NAIP-/-, and CASP1-/- THP-1 cells using a chloroquine (CHQ) resistance assay. This data has now been included in the manuscript in the new Figure 5A. The original subfigures of Figure 5 have consequently been rearranged. We did not observe any significant differences in vacuolar and cytosolic bacterial burdens at this early time point in WT, NAIP-/-, and CASP1-/- THP-1 cells. As noted by the Reviewer, these results suggest that the basal level of bacterialdependent lysis of the SCV in human macrophages is not dependent on caspase-1 or NAIP.

**Reviewer #3:**
(1) The main weaknesses of the study are the inherent limitations of tissue culture models. For example, to study interaction of *Salmonella* with host cells in vitro, it is necessary to kill extracellular bacteria using gentamicin. However, since *Salmonella*-induced macrophage cell death damages the cytosolic membrane, gentamicin can reach intracellular bacteria and contribute to changes in CFU observed in tissue culture models (major point 1). This can result in tissue culture "artefacts" (i.e., observations/conclusions that cannot be recapitulated in vivo). For example, intracellular replication of *Salmonella* in murine macrophages requires T3SS-2 in vitro, but T3SS-2 is dispensable for replication in macrophages of the spleen in vivo (Grant et al., 2012).

We thank the Reviewer for their helpful comments and insightful suggestions. We have addressed some of the concerns about gentamicin in our response to Reviewer #1 above. To address the Reviewer’s concerns further, we have included language to acknowledge the limitations of our study based on the artefacts of tissue culture models in our Discussion section: “In this study, we utilized tissue culture models to examine intracellular *Salmonella* replication in human macrophages. These in vitro systems allow for precise control of experimental conditions and, therefore, serve as powerful tools to interrogate the molecular mechanisms underlying inflammasome responses and *Salmonella* replication in both immortalized and primary human cells. Still, there are limitations of tissue culture models, as they lack the inherent complexity of tissues and organs in vivo. To assess whether our findings reflect *Salmonella* dynamics in the mammalian host, it will be important to complement our studies and extend the implications of our work using approaches that model more complex systems, such as organoids or organ explant models co-cultured with immune cells, and in vivo techniques, such as humanized mouse models.”

(2) In Figure 1: are increased CFU in WT vs CASP1-deficient THP-1 cells due to Caspase 1 restricting intracellular replication or due to Caspase-1 causing pore formation to allow gentamicin to enter the cytosol thereby restricting bacterial replication? The same question arises about Caspase-4 in Figure 2, where differences in CFU are observed only at 24h when differences in cell death also become apparent. The idea that gentamicin entering the cytosol through pores is responsible for controlling intracellular *Salmonella* replication is also consistent with the finding that GSDMD-mediated pore formation is required for restricting intracellular *Salmonella* replication (Figure 3). Similarly, the finding that inflammasome responses primarily control *Salmonella* replication in the cytosol could be explained by an intact SCV membrane protecting *Salmonella* from gentamicin (Figure 5).

We thank the Reviewer for highlighting this important point regarding gentamicin.

We have addressed this question in our response above to Review #1 and in Figure 1 – Figure Supplement 5. We observed caspase-1-mediated restriction of *Salmonella* in human macrophages even when cells were treated with a lower concentration of gentamicin (25 μg/ml) for 30 minutes and then extensively washed with RPMI media to remove any gentamicin for the remainder of the infection. These data suggest that gentamicin is likely not responsible for controlling intracellular *Salmonella* in human macrophages.